# A molecular portrait of maternal sepsis from Byzantine Troy

**Alison M Devault[1,2†], Tatum D Mortimer[3,4†], Andrew Kitchen[5], Henrike Kiesewetter[6], Jacob M Enk[1,2], G Brian Golding[7], John Southon[8], Melanie Kuch[1], Ana T Duggan[1], William Aylward[9,10], Shea N Gardner[11], Jonathan E Allen[11], Andrew M King[12], Gerard Wright[12], Makoto Kuroda[13], Kengo Kato[13], Derek EG Briggs[14], Gino Fornaciari[15], Edward C Holmes[16], Hendrik N Poinar[1,7,12,17]\*, Caitlin S Pepperell[3,9,18]\***

[1]McMaster Ancient DNA Centre, Department of Anthropology, McMaster University, Hamilton, Canada; [2]MYcroarray, Ann Arbor, United States; [3]Department of Medical Microbiology and Immunology, School of Medicine and Public Health, University of Wisconsin-Madison, Madison, United States; [4]Microbiology Doctoral Training Program, University of Wisconsin-Madison, Madison, United States; [5]Department of Anthropology, University of Iowa, Iowa City, United States; [6]Project Troia, Institute of Prehistory, Early History, and Medieval Archaeology, Tübingen University, Tübingen, Germany; [7]Department of Biology, McMaster University, Hamilton, Canada; [8]Keck Carbon Cycle Accelerator Mass Spectrometer, Earth Systems Science Department, University of California, Irvine, United States; [9]Molecular Archaeology Laboratory, Biotechnology Center, University of Wisconsin-Madison, Madison, United States; [10]Department of Classics and Ancient Near Eastern Studies, University of Wisconsin-Madison, Madison, United States; [11]Lawrence Livermore National Laboratory, Livermore, United States; [12]Michael G. DeGroote Institute for Infectious Disease Research, McMaster University, Hamilton, Canada; [13]Laboratory of Bacterial Genomics, Pathogen Genomics Center, National Institute of Infectious Diseases, Tokyo, Japan; [14]Department of Geology and Geophysics, Yale University, New Haven, United States; [15]Division of Paleopathology, Department of Translational Research on New Technologies in Medicine and Surgery, University of Pisa, Pisa, Italy; [16]Marie Bashir Institute for Infectious Diseases and Biosecurity, Charles Perkins Centre, School of Life and Environmental Sciences and Sydney Medical School, The University of Sydney, Sydney, Australia; [17]Humans and the Microbiome Program, Canadian Institute for Advanced Research, Toronto, Canada; [18]Department of Medicine (Infectious Diseases), School of Medicine and Public Health, University of Wisconsin-Madison, Madison, United States

\*For correspondence: poinarh@ mcmaster.ca (HNP); cspepper@ medicine.wisc.edu (CSP)

†These authors contributed equally to this work

**Competing interests:** The authors declare that no competing interests exist.

**Abstract** Pregnancy complications are poorly represented in the archeological record, despite their importance in contemporary and ancient societies. While excavating a Byzantine cemetery in Troy, we discovered calcified abscesses among a woman's remains. Scanning electron microscopy of the tissue revealed 'ghost cells', resulting from dystrophic calcification, which preserved ancient maternal, fetal and bacterial DNA of a severe infection, likely chorioamnionitis. *Gardnerella vaginalis* and *Staphylococcus saprophyticus* dominated the abscesses. Phylogenomic analyses of ancient, historical, and contemporary data showed that *G. vaginalis* Troy fell within contemporary genetic diversity, whereas *S. saprophyticus* Troy belongs to a lineage that does not appear to be

commonly associated with human disease today. We speculate that the ecology of *S. saprophyticus* infection may have differed in the ancient world as a result of close contacts between humans and domesticated animals. These results highlight the complex and dynamic interactions with our microbial milieu that underlie severe maternal infections.

## Introduction

During excavations of a Late Byzantine era cemetery at the periphery of the ancient city of Troy, Anatolia (in present day Turkey) (*Figure 1—figure supplement 1*), we discovered two calcified nodules among a woman's remains. The woman was estimated to be 30 (±5y) at the time of death (Appendix). She was found alone in a stone-lined grave (*Figure 1A*) within the graveyard of a farming community (*Kiesewetter, 2014*). The nodules, which are 2–3 cm in diameter and composed of concentric layers (*Figure 1B*), were discovered at the base of the ribs. Radiocarbon dating of the decedent's ulna yielded 790-860y BP (*Supplementary file 1A*), in agreement with the archeological assessment of the age of the cemetery (early 13th century AD, Appendix).

Nodule one (*Figure 2—figure supplement 1*, *Supplementary file 1B*) is primarily composed of two phosphate phases, hydroxylapatite (bioapatite) and whitlockite (as well as small amounts of calcite), both of which have been found in human calcified pathological concretions (*Lagier and Baud, 2003*). Based on their size and concentric layered structure, the nodules could be urinary stones. However, struvite (magnesium ammonium phosphate) and calcium oxalate, common constituents of urinary stones, were absent in both XRD and SEM-EDS analyses (*Supplementary file 1B-D*). SEM of the nodules (*Figure 2*, *Figure 2—figure supplement 2*) revealed aggregates of spherical structures with dimensions typical of bacterial cells, as well as extracellular polymeric substances (EPS – a glycocalyx secreted by the cells during biofilm formation [*Decho and Thiel, 2011*]).

We extracted DNA from both nodules and made Uracil DNA Glycosylase (UDG) and non-UDG treated dsDNA libraries. Shotgun sequences from all libraries yielded astonishingly high proportions of endogenous human and bacterial DNA: 24–48% human, 37–66% *S. saprophyticus*, and 5–7% *G. vaginalis* (*Figure 1—figure supplements 2–5*).

From these data, we reconstructed a human mitochondrial genome at 30.1x unique read depth, the consensus of which belongs to haplotype U3b3. In phylogenetic analyses of the mitogenome from Troy and modern mitogenomes, the Troy sample groups most closely with those from the Caucasus and Middle East, both of which were within the eastern limits of Late Byzantine influence (*Figure 1—figure supplement 6*).

To investigate whether the nodules belonged to the associated female individual, we extracted DNA from her ulna, constructed a dsDNA library, enriched for, sequenced, and reconstructed the mitogenome to an average unique coverage depth of 30.8x. The nodule and the ulna share the identical mitochondrial haplotype (*Supplementary file 1E*), indicating that they stem from the same individual or a maternal relative.

The metagenomic profile of the nodules suggests they derive from an amalgam of human and bacterial cells, as in an abscess. The high concentration of *S. saprophyticus* and *G. vaginalis* DNA suggests an origin in genitourinary tissue. To exclude an exogenous environmental source of the bacterial DNA and to further investigate the tissue of origin, we performed metagenomic profiling of the nodules, ulna and sediment from the gravesite. The similarity in abundance of *G. vaginalis* in the nodules and modern Human Microbiome Panel (HMP) vaginal samples (*Figure 1—figure supplement 7*) points to a likely origin for the nodules in the female reproductive tract. The metagenomic profile of the nodules (minus their associated blanks) is distinct from the sediment, whereas the reads from the ulna group closely with the sediment sample (*Figure 1—figure supplement 8*). These results indicate that the nodules were less prone to leaching of environmental DNA. Our SEM-EDS and XRD findings suggest that bacterial and inflammatory cells were replicated in calcium phosphate minerals ('ghost cells'); it is likely that this mineralization provided a remarkable degree of protection from DNA degradation and environmental leaching as seen in the bones. The ectopic, inflammation-related calcification observed here is an apparently highly effective mechanism of bacterial fossilization that rivals mineralization occurring at much slower rates in the environment.

**eLife digest** Why and how have some bacteria evolved to cause illness in humans? One way to study bacterial evolution is to search for ancient samples of bacteria and use DNA sequencing technology to investigate how modern bacteria have changed from their ancestors. Understanding the evolution process may help researchers to understand how some bacteria become resistant to the antibiotics designed to kill them.

Complications that occur during pregnancy, including bacterial infections, have long been a major cause of death for women. Now, Devault, Mortimer et al. have been able to sequence the DNA of bacteria found in tissue collected from a woman buried 800 years ago in a cemetery in Troy. Some of the woman's tissues had been well preserved because they had calcified (probably as the result of infection), which preserved their structure in a mineralized layer. Two mineralized "nodules" in the body appear to be the remains of abscesses. Some of the human DNA in the nodules came from a male, suggesting that the woman was pregnant with a boy and that the abscesses formed in placental tissue.

Sequencing the DNA of the bacteria in the abscess allowed Devault, Mortimer et al. to diagnose the woman's infection, which was caused by two types of bacteria. One species, called *Gardnerella vaginalis*, is found in modern pregnancy-related infections. The DNA of the ancient samples was similar to that of modern bacteria. The other bacteria species was an ancient form of *Staphylococcus saprophyticus*, a type of bacteria that causes urinary tract infections. However, the DNA of the ancient *S. saprophyticus* bacteria is quite different to that of the bacteria found in modern humans. Instead, their DNA sequence appears more similar to forms of the bacteria that infect currently livestock. As humans lived closely with their livestock at the time the woman lived, her infection may be due to a type of bacteria that passed easily between humans and animals.

Overall, the results suggest that the disease-causing properties of bacteria can arise from a wide range of sources. In addition, Devault, Mortimer et al. have demonstrated that certain types of tissue found in archeological remains are a potential gold mine of information about the evolution of bacteria and other microbes found in the human body.

Sexing analyses of the remains (and associated blanks) using the method of *Skoglund et al. (2013)* assign the nodules as female -XX (*Supplementary file 1F*). More thorough analyses of the human DNA present in the nodules yielded an intriguing finding that helps pinpoint their tissue of origin. Shotgun sequencing data from the nodules contained a small number of reads (884) conservatively mapping to the Y chromosome (*Supplementary file 1G*). The length distributions of the reads overlapped with those mapping to the X chromosome and autosomes, suggesting an endogenous, ancient origin (*Supplementary file 1G*, *Figure 1—figure supplement 5*); we searched for but did not find similar *bona fide* ancient Y chromosome reads in sequence data from the ulna, the sediment, or any negative control (*Supplementary file 1G*). The presence of Y chromosome reads in the nodule but not in the ulna could be explained by a placental origin of the mineralized abscesses, indicating chorioamnionitis in the decedent while pregnant with a male fetus. Chorioamnionitis – inflammation and infection of the placenta and fetal membranes – involves an inflammatory response on the part of the fetus as well as the mother (*Kraus et al., 2004*), which would explain a female (maternal) origin of the nodular tissue with a minority male (fetal) component.

Chorioamnionitis is a mixed infection in which vaginal bacteria reach the upper reproductive tract, placenta, and fetal membranes; *G. vaginalis* is often identified in infected tissues (*Hillier et al., 1988*). *S. saprophyticus* can be found among the genitourinary and gastrointestinal flora of healthy women (*Ringertz and Torssander, 1986*; *Rupp et al., 1992*; *Schneider and Riley, 1996*). It is a common cause of urinary tract infection (UTI) in reproductive aged women (*Kahlmeter, 2003*) and has also been known to cause puerperal infections (*Arianpour et al., 2009*).

To gain further insights into the pathogens associated with this historical genitourinary infection, we pooled reads from all nodule DNA libraries, mapped and reconstructed the ancient *S. saprophyticus* and *G. vaginalis* genomes and analyzed them in conjunction with existing and newly acquired genomic data from extant and historical organisms (*Supplementary file 1H,I*).

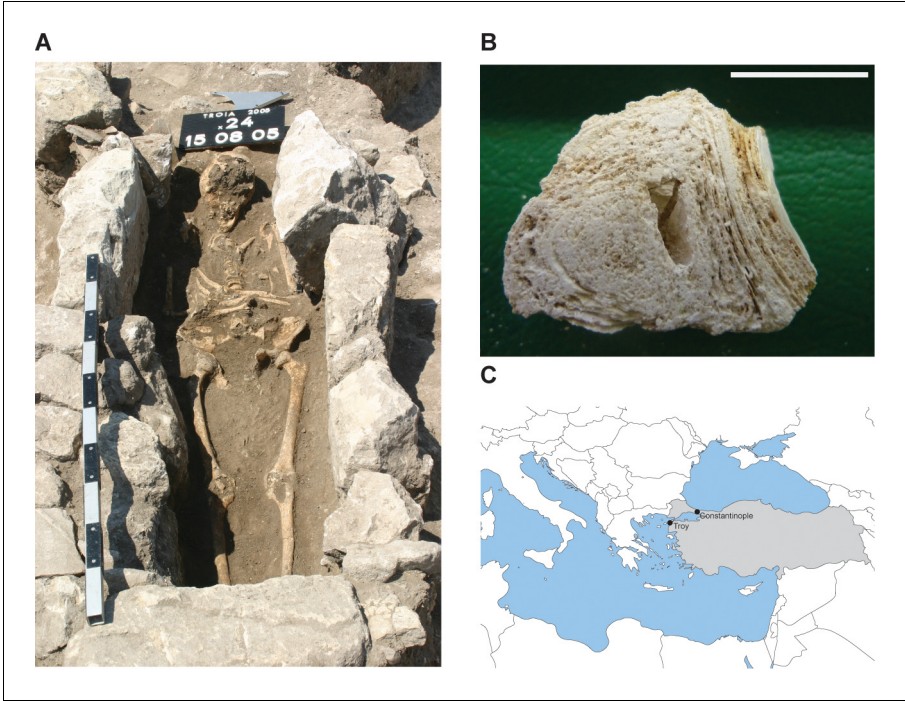

**Figure 1.** Calcified nodule found among the skeletal remains at Troy. (**A**) Burial x24.177 (grave 14, cemetery in quadrat x24). Photo credit Gebhard Bieg, 2005. (**B**) Cross-section of nodule (sample no x24.177), photo credit: Pathologie Nordhessen 2009. Scale represents 1 cm. (**C**) Location of Troy. Modern day Turkey is shaded in gray.

The following figure supplements are available for figure 1:

**Figure supplement 1.** Map of Troy showing the cemetery in Grid Square x24 and areas of excavation 1988–2012.

**Figure supplement 2.** Metagenomic profiles of shotgun DNA libraries from nodules, based on BLAST analysis of all reads >35 bp length.

**Figure supplement 3.** Fragment length distributions for non-UDG treated human mitochondrial assemblies.

**Figure supplement 4.** Ancient DNA damage assessment of human mitochondrial reads.

**Figure supplement 5.** Ancient DNA damage assessment of reads mapped to hg38 chrX, chrY and autosomes.

**Figure supplement 6.** Haplogroup U3 Bayesian Maximum Clade Credibility tree.

**Figure supplement 7.** Heatmap of most common taxa in metagenomic samples.

**Figure supplement 8.** PCA of Human Microbiome Project and ancient metagenomic taxa.

**Figure supplement 9.** Sketch of skeletal preservation.

We used a combination of paired-end reference guided assembly and iterative assembly to reconstruct a nearly complete genome of *S. saprophyticus* Troy, including >100 Kb of novel sequence compared to reference strain ATCC 15305. The genome is 2,471,881 bp long, with an average unique coverage depth of 298.6x (*Figure 4—figure supplement 3*), which represents an unprecedented, detailed and complete picture of an ancient pathogen genome from shotgun sequencing data. We also reconstructed a 22.6 Kb plasmid, pSST1.

We were unable to reconstruct a contiguous *G. vaginalis* genome due to high variability in coverage and lack of synteny in both ancient and modern genomic data (*Ahmed et al., 2012*). Instead,

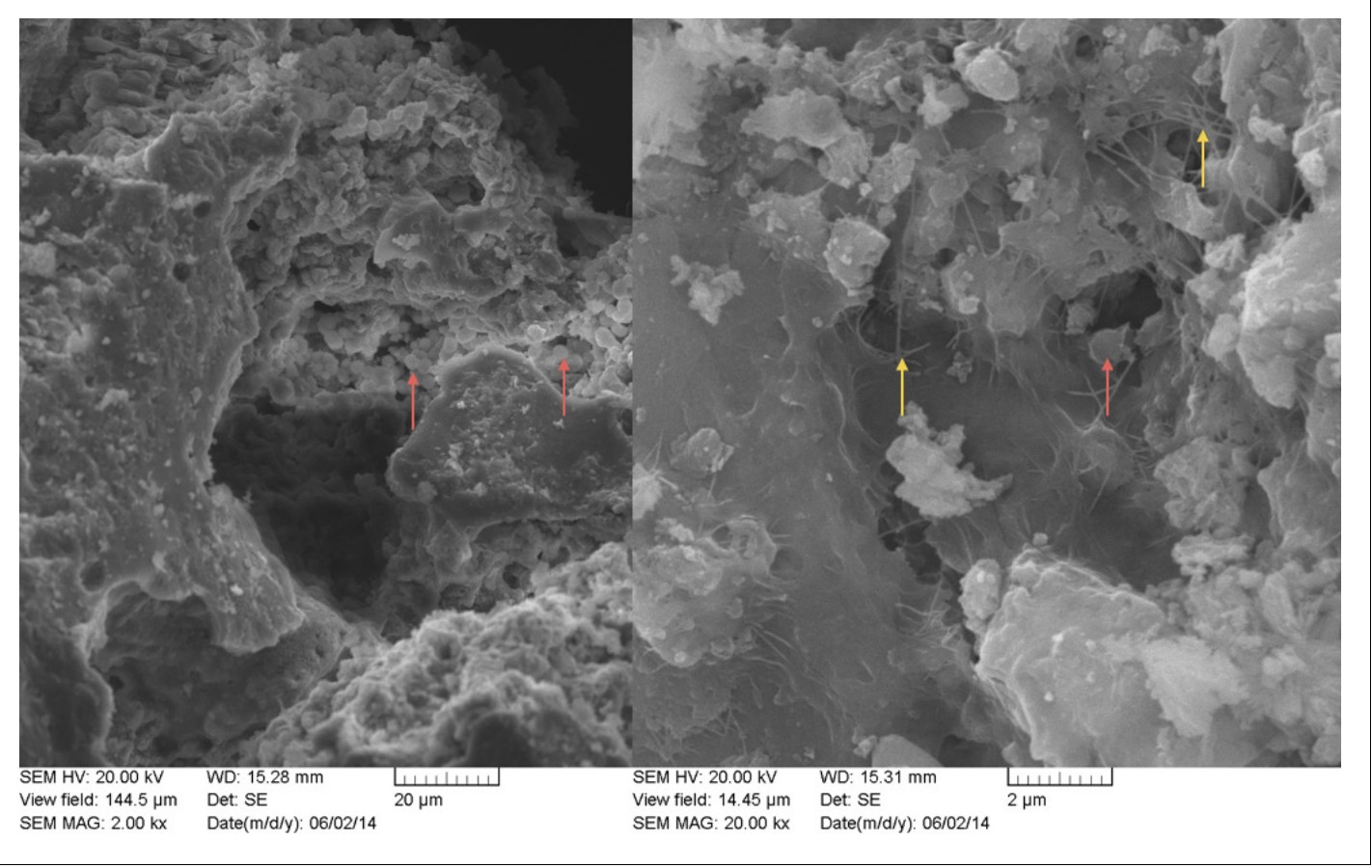

**Figure 2.** SEM image of nodule at (**A**) 2000x and (**B**) 20,000x magnification. Bacterial cells indicated with red arrow are between ~1 µm and 2 µm (within range expected for *Staphylococcus*). Extracellular polymeric substances (EPS) are indicated by yellow arrows.

The following figure supplements are available for figure 2:

**Figure supplement 1.** XRD analysis of nodule.

**Figure supplement 2.** SEM image of nodule at 10,000x magnification.

we used a *de novo* approach to reconstruct *G. vaginalis* Troy gene content using reads that mapped to the annotated coding regions of all available *G. vaginalis* genomes. This enabled us to assess the gene content of our ancient genome compared to the modern strains. Using this method, we recovered 1187 unique contigs (total length 1,435,761 bp) corresponding to 972 annotated genes and an average unique coverage depth of 57.0x (*Figure 3—figure supplement 3*).

Our sample of 35 isolates of *G. vaginalis* was grouped into four previously defined clades (*Figure 3*, *Figure 3—figure supplement 4*), which have been proposed to represent distinct species (*Ahmed et al., 2012*). *G. vaginalis* Troy sits within Clade 1, among vaginal and endometrial isolates collected from both healthy women and patients with bacterial vaginosis. Interestingly, the 800-year-old sample from Troy (Turkey) falls within contemporary genetic diversity (*Supplementary file 1I*).

Consistent with prior reports (*Ahmed et al., 2012*), we identified extensive impacts of lateral gene transfer (LGT) on *G. vaginalis* diversity (*Figure 3*). Even in the core genome alignment, which contains just 44% of per-isolate gene content, we estimate that 20% of sites have been affected by recombination. This high rate of recombination may help to explain the remarkable preservation of genetic diversity in *G. vaginalis*. A recent study of *Helicobacter pylori*, which has similarly high rates of LGT, found that genetic diversity within the species has been preserved for more than five thousand years (*Maixner et al., 2016*).

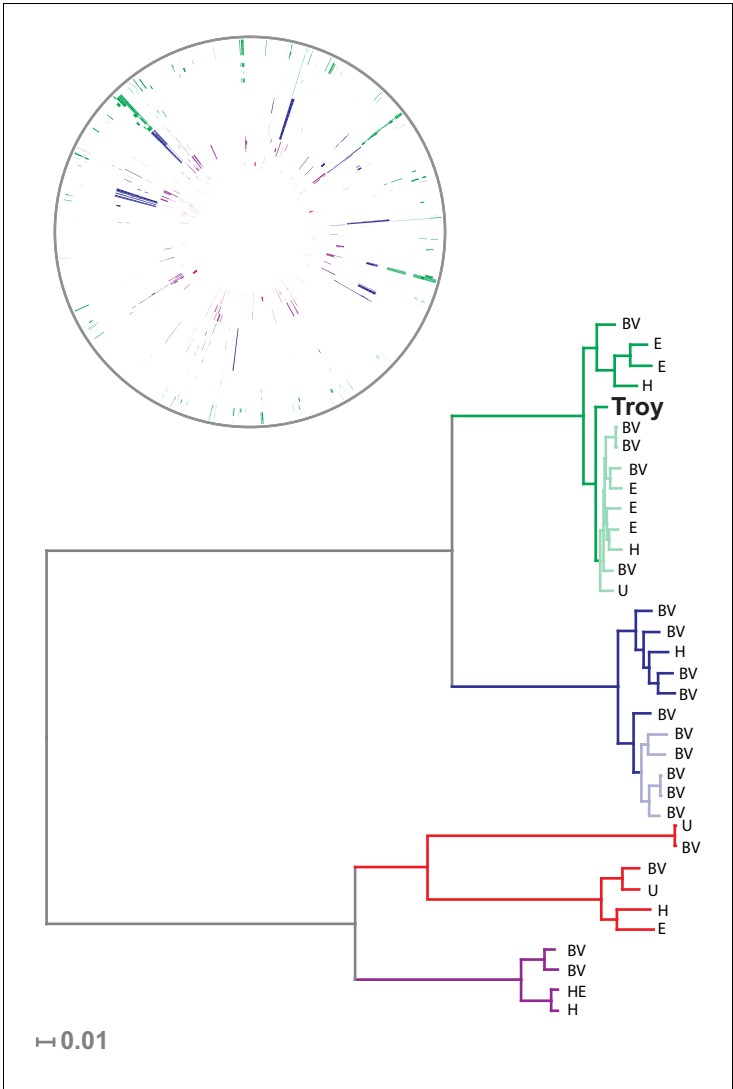

**Figure 3.** Phylogenetic analysis of *Gardnerella vaginalis*. A maximum likelihood tree estimated using RAxML (*Stamatakis, 2014*) (*Figure 3—source data 3*) from a core alignment of *G. vaginalis* genomes (*Figure 3—source data 1*, *Figure 3—source data 2*). Branches are colored based on clades originally identified in Ahmed et al. (*Ahmed et al., 2012*) (green = clade 1, blue = clade 2, red = clade 3, purple = clade 4). Tips from modern *G. vaginalis* isolates are labeled based on sample source (H = healthy vagina, BV = bacterial vaginosis, HE = healthy endometrium, E = endometrium, U = unknown). Lighter colored branches have bootstrap values less than 100. Clinical phenotypes are interspersed throughout the phylogeny, and the Troy genome is not associated with a consistently pathogenic lineage of *G. vaginalis*. Inset: Recombinant fragments in *G. vaginalis* core genome identified by BratNextGen (*Figure 3—source data 4*) (*Marttinen, 2012*). Each circle represents one genome. Colored blocks represent recombinant fragments, and colors correspond to the clade designations in the phylogenetic tree. Plot made with Circos (*Krzywinski et al., 2009*).

The following source data and figure supplements are available for figure 3:

**Source data 1.** Concatenated alignment of core genes in *G. vaginalis*.

**Source data 2.** *G. vaginalis* core genome alignment trimmed with Gblocks.

**Source data 3.** Maximum likelihood phylogenetic analysis of trimmed *G. vaginalis* alignment with RAxML.

**Source data 4.** Recombinant fragments detected with BratNextGen in trimmed *G. vaginalis* alignment.

*Figure 3 continued*

**Figure supplement 1.** Ancient DNA damage assessment of *G. vaginalis.*
**Figure supplement 2.** Fragment length distribution (FLD) for *G. vaginalis* ATCC 14019.
**Figure supplement 3.** Genome coverage plots for pooled nodule shotgun libraries.
**Figure supplement 4.** Neighbor net network of core genomes.

We discovered two distinct clades of *S. saprophyticus* (*Figure 4*, *Figure 4—figure supplement 4*), one of which (Clade P) appears to be more strongly associated with pathogenicity and includes our ancient *S. saprophyticus* Troy. Nineteen of twenty veterinary and human clinical isolates belong to Clade P, an association that was statistically significant (Appendix). A second clade (Clade E) is made up of food and environmental isolates of *S. saprophyticus*, as well as a human UTI isolate from Japan.

Plasmids similar to *S. saprophyticus* Troy pSST1 were present in isolates from both clades. The relationships among plasmid sequences from *S. saprophyticus* Troy and other members of Clade P were distinct from those of the core genome; we also found evidence of recombination among pSST1-like plasmids (*Figure 4C*, Appendix).

A long branch separates Clade P pSST1 from those of Clade E (*Figure 4C*), recapitulating their relationship on the core genome phylogeny. This was also true of pSSP2, the only other plasmid we identified in both Clades P and E (but not *S. saprophyticus* Troy; *Figure 4—figure supplement 5*). These observations suggest plasmids are more readily exchanged within Clades P and E than between them. This could indicate that Clades P and E are spatially segregated, that there are mechanistic barriers to plasmid exchange between clades, or that epistatic interactions reinforce clade separation of these mobile elements.

The human clinical isolates in Clades P and E are nested within the phylogeny with more divergent lineages associated with other animals. This suggests that the most recent common ancestor (MRCA) of *S. saprophyticus* may not have been human-associated. This is in stark contrast to the major pathogen in the genus, *Staphylococcus aureus*, where phylogenetic studies suggest that the MRCA of human and livestock-associated lineages had a human host (*Fitzgerald, 2012*; *Weinert et al., 2012*; *Shepheard et al., 2013*). *S. aureus* is strongly associated with its niche on the human body and is transmitted primarily from person-to-person. *S. saprophyticus*, by contrast, appears to be a generalist that colonizes a range of environments.

Several lines of evidence also indicate that humans acquire *S. saprophyticus* infection from the environment. In northern climates, there is marked seasonal variation in the incidence of *S. saprophyticus* UTI (*Rupp et al., 1992*; *Hovelius and Mårdh, 1984*; *Ringertz and Torssander, 1986*; *Hedman et al., 1993*; *Widerström et al., 2007*), whereas there is no evidence of seasonality in Mediterranean climates (*Schneider and Riley, 1996*). *S. saprophyticus* can be identified in environmental samples, with a strong seasonal peak that occurs just before peak rates of *S. saprophyticus* UTI in northern climates (*Hedman et al., 1993*; *Soge et al., 2009*). Molecular epidemiological surveys also suggest *S. saprophyticus* is primarily acquired from an environmental reservoir, rather than as a result of person-to-person transmission (*Widerström et al., 2007*; *Widerström et al., 2012*). These observations suggest that the bacteria cycle between host-associated and environmental stages, with seasonal climatic effects on their abundance in the environment.

The length of the branch leading to *S. saprophyticus* Troy is similar to those leading to the other tips (*Figure 4A*), suggesting there is little temporal signal in the phylogeny. Calibrated phylogenetic analyses (Appendix) confirmed the absence of temporal signal, which precludes reliable estimation of the rate of substitution or divergence times for *S. saprophyticus*.

A mixed environmental, commensal and pathogenic niche may in part explain the absence of temporal structure in our sample of *S. saprophyticus*. Selection pressures and generation times are likely to differ between free-living and host-associated stages, which can obscure temporal signals in genetic data (*Bromham, 2009*). In addition to producing rate variability, periods of dormancy in the environment – e.g. during the winter in northern climates, as suggested by seasonal patterns in cultivability – would be predicted to lower the overall rate at which *S. saprophyticus* evolves

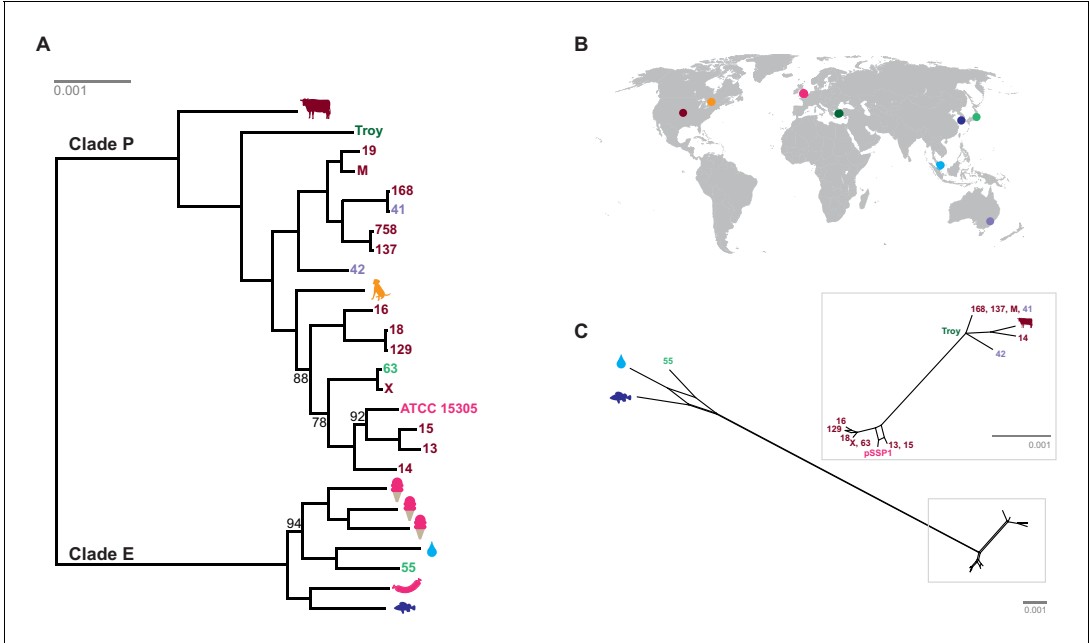

**Figure 4.** Phylogenetic analysis of *Staphylococcus saprophyticus*. (**A**) Maximum likelihood tree estimated using RAxML (***Stamatakis, 2014***) (***Figure 4—source data 3***) from an alignment of *S. saprophyticus* genomes (***Figure 4—source data 1***, ***Figure 4—source data 2***). Bootstrap values less than 100 are labeled. Silhouettes indicate bacterial sample source. Isolates without silhouettes are from human clinical samples isolated from urine. Color corresponds to country of isolation as seen on the map. Full sample descriptions are in ***Supplementary file 1H***. (**B**) Source countries of bacterial samples. (**C**) Neighbor-net network of *S. saprophyticus* plasmid sequences (***Figure 4—source data 4***) related to pSST1 created in SplitsTree4 (***Huson and Bryant, 2006***). The boxed inset is an enlarged version of the portion of the network from Clade P isolates. Some *S. saprophyticus* isolates do not encode pSST1-like plasmids, and therefore, they are not included in the network. Starts and stops of recombinant regions of the alignment can be found in ***Figure 4—source data 5***.

The following source data and figure supplements are available for figure 4:

**Source data 1.** *S. saprophyticus* whole genome alignment.
**Source data 2.** *S. saprophyticus* whole genome alignment trimmed with trimal.
**Source data 3.** Maximum likelihood phylogenetic analysis of trimmed *S. saprophyticus* alignment with RAxML.
**Source data 4.** *S. saprophyticus* plasmid alignment trimmed with trimal.
**Source data 5.** Recombinant fragments detected with BratNextGen in *S. saprophyticus* alignment.
**Figure supplement 1.** Ancient DNA damage assessment of *S. saprophyticus*.
**Figure supplement 2.** Fragment length distribution (FLD) for *S. saprophyticus* ATCC 15305.
**Figure supplement 3.** Genome coverage plots for pooled nodule shotgun libraries.
**Figure supplement 4.** Neighbor net network of core genomes.
**Figure supplement 5.** Presence of mobile genetic elements, virulence genes, and antibiotic resistance in *S. saprophyticus*.
**Figure supplement 6.** Recombinant regions detected by BratNextGen in *S. saprophyticus*.

(***Bromham, 2009***; ***Weinert et al., 2015***). The 800 year interval between *S. saprophyticus* Troy and the other tips may simply be too short relative to the overall depth of the tree to allow reliable rate inference.

Notably, all human-associated isolates of *S. saprophyticus* in Clade P form a monophyletic group, to which the bovine mastitis strain falls basally; there are no modern human pathogenic representatives of the *S. saprophyticus* Troy lineage. This may mean that the ecology of *S. saprophyticus* differed in the Byzantine world, with human infections arising from a different reservoir of bacteria than they do today. *S. saprophyticus* is readily cultured from the environment around livestock (*Hedman et al., 1993*; *Cherif-Antar et al., 2016*), and Byzantine era peasants in Anatolia typically shared their households with cattle (*Lefort, 2007*). This and other historical settings are likely to have facilitated spillover events and, perhaps, the circulation of bacteria that were adapted to both livestock and humans.

Based on the available data, it is not possible to determine whether the human clinical isolate nested among environmental and food-associated bacteria in Clade E represents a spillover or a second emergence into humans. In either event, it appears that *S. saprophyticus* can transition to a human pathogenic niche with relative ease. We did not identify any gene content uniquely shared (or absent) among the pathogenic strains in our sample, which suggests that pleiotropy underlies *S. saprophyticus'* flexible association with diverse niches. For many bacterial genera, genetic distances between free-living organisms and pathogens are larger than observed here, and pathogen emergence is a singular event characterized by genomic decay and loss of functions required outside the pathogenic niche (*Parkhill et al., 2001*; *Larsson et al., 2009*; *Reuter et al., 2014*). More studies and wider sampling are needed to fully characterize the niche of *S. saprophyticus,* but our observations reinforce the notion that the adaptive paths to bacterial virulence are more diverse than has previously been appreciated.

Complications of pregnancy and childbirth are major causes of morbidity and mortality worldwide and new threats to maternal health continue to emerge (*WHO et al., 2014*; *Mlakar et al., 2016*). Our analyses of the remains of a woman who died in Late Byzantine Troy connect her to this broad historical and epidemiological phenomenon. Her infection was associated with exuberant calcification of the placenta, which replicated maternal, fetal, and bacterial cells in calcium phosphate minerals and preserved a high resolution molecular portrait of their contents. *S. saprophyticus*, an organism the decedent is likely to have acquired from her environment, and *G. vaginalis*, a member of the native human biota, are the dominant bacterial species of the infection. *S. saprophyticus* Troy belongs to a lineage that appears to be uncommonly associated with human disease in the modern world, whereas *G. vaginalis* Troy nests among modern commensal and pathogenic strains on its phylogeny. This highlights the complexity of virulence as a bacterial trait and a potential role of interactions among bacterial species in shaping pathologic outcomes of infection.

## Materials and methods

### Samples

Ethics approval for the study of the remains of the individual excavated in 2005 from grave 14 (Troy project, University of Tübingen, bone-sample x24.177) in quadrat x24 at Troy was obtained from Hamilton Health Sciences and McMaster University (REB# 13–146 T). Samples of extant bacteria were provided to investigators without patient identifiers or protected health information; the members of the study team did not have access to any identifiers or protected health information associated with the bacterial isolates. Sediment from the Troy site was imported to and studied at McMaster University in accordance with Canadian Food Inspection Agency guidelines, under permit number P-2012–04220.

### Radiocarbon dating

Subsamples of both nodules and the ulna bone were sent to the Keck Carbon Cycle AMS Facility (Earth System Science Department, UC Irvine, CA) for radiocarbon dating (Appendix, *Supplementary file 1A*). In addition to $^{14}$C dating ultrafiltered collagen from the ulna, we also attempted to measure $^{14}$C in carbonate from nodule one (with 10–30% leaching) and three organic fractions from nodule two: raw nodule (including carbonate), residue from demineralization with room temperature 1N HCl, and residue from demineralization plus gelatinization with 60°C 0.01N HCl.

## X-ray diffraction (XRD)

A subsample of nodule two was subjected to mineralogical analysis using XRD at the Brockhouse Institute for Materials Research (McMaster University) using the Bruker D8 DISCOVER with DAVINCI. DESIGN diffractometer (*Figure 2—figure supplement 1*). Sample flakes were piled on top of a single crystal silicon wafer, and aligned to the center of the diffractometer using the laser-video alignment system. The detector to sample distance was calibrated with corundum to 20 cm. Four frames were collected to obtain the 2θ range of 8–103°. The frames were integrated into intensity plots using DIFFRAC.EVA V.3.0 (software package from Bruker AXS). A pattern search/match was executed using the integrated ICDD PDF-2 2011 powder database. Slight mismatch in the peak positions are likely due to variation of elemental stoichiometry in the identified phase.

## Scanning electron microscopy (SEM) with energy dispersive X-ray spectroscopy (EDS)

A subsample of nodule two was viewed via SEM and subjected to elemental analysis using SEM-EDS at the MAX Diffraction Facility (McMaster University). Sample pieces were attached to an aluminum stub with double-sided carbon tape and sputter-coated with a thin layer of Au. The sample was viewed in a Tescan Vega II LSU operating at 20kV. Energy dispersive spectroscopy (EDS) was carried out with an IMAX detector (Oxford Instruments) and INCA software (*Supplementary file 1B*).

## Ancient DNA extractions

We made multiple DNA extractions from subsamples of two nodules, an ulna, sediment taken from the site and relevant associated blanks/controls. The details of these can all be found in *Supplementary file 1J*. As the elemental analyses of the nodules suggested a highly mineralized constituent, we extracted them in a similar fashion to bone and they are labelled as such in *Supplementary file 1J* hyphenated ('bone'). 'Bone' (nodules) DNA extractions, consisted of demineralization (DM), removal and freezing of DM supernatant, incubation of non-demineralized tissue with a custom digestion buffer (DB), removal and freezing of DB supernatant, organic extraction of one/both supernatants, and concentration via filtration. They were performed as follows. For Set A (*Supplementary file 1J*), multiple consecutive rounds of DM (with 0.5M EDTA) and digestion were performed on a shaker (1000 rpm), with the supernatant(s) from each round subjected to organic extraction and filtration. DB consisted of 20 mM Tris pH 8.0, 0.5% sarcosyl, 250 µg/ml Proteinase K, 5 mM CaCl2, 50 mM DTT, 1% PVP, and 2.5 mM PTB. The breakdown of DM/DB rounds is as follows: round 1 = 1 st 0.5 mL overnight (ON) DM at room temperature (RT), second 0.5 mL DM at RT for 24 hr, and 7 hr digestion at 55°C, rounds 2–5 = ON 1 mL DM + 7 hr digestion at 55°C, and round 6 = ON 1 mL DM only. For subsequent extraction, all round one supernatants were combined and all round 2–6 supernatants were treated separately: supernatants were subjected to organic extraction using half-volume of phenol-chloroform-isopropanol (centrifuged at 16,000 x g for 5m), the aqueous phase of which was extracted with 750 µl chloroform (centrifuged as before). The final aqueous phase was filtered using Amicon Ultra 0.5 mL 10 kDa columns (EMD Millipore Corp., Billerica, MA, USA): columns primed with 450 µl 0.1xTE, followed by sample filtration, washed twice with 450 µl 0.1xTE, and eluted in 50 µl 0.1xTE. For Set B, nodule two, 1 mL DM was performed ON rotating at RT and the supernatant was frozen. Digestion was performed for 7 hr rotating at 55°C using 0.5 mL of DB (same recipe as Set A) and the supernatant was frozen. These supernatants were combined and subjected to organic extraction and filtration as in Set A. Set D, which was the ulna, followed the same protocol as Set B, except they were subjected to an additional initial 30 min demineralization with 500 µl 0.5M EDTA that was not extracted, and the DB did not contain DTT, PVP, or PTB and was performed ON.

Set C, 'sediment', DNA extractions were performed using the Mo Bio PowerSoil DNA Isolation Kit (MO BIO Laboratories, Inc., Carlsbad, CA) following the manufacturer's protocol, with a final elution of 100 µl in 0.1xTE. For each set of samples associated blanks were treated in an identical fashion. Please refer to *Supplementary file 1J* for details for each of these four sets of extractions.

## Library preparation and indexing

Ancient DNA extracts were converted to double-indexed libraries for sequencing on the Illumina platform (list in *Supplementary file 1A*). Prior to library preparation, all ulna DNA extracts (E5-1, 2,

3, 4, 7, 8, 9, and 10) were pooled to homogenize library input into multiple UDG and non-UDG libraries, as were the two set D extraction blanks (E5-6 and 12). Libraries were prepared as in (*Wagner et al., 2014*) using either regular ('non-UDG') or deaminated cytosine removal (damage repair; 'UDG') protocols, and subsequently amplified using a double-indexing protocol (*Kircher et al., 2012*; *Meyer and Kircher, 2010*). Each library set included at least one blank no-template (water) control reaction. Extract input volumes into library preparation were 10 µl (L25-L13), 20 µl (L01-L20), or 25 µl (L28-L38, 1 a-1j, and 1a-blk to 1j-blk). In non-UDG libraries L25-L38, the blunt-ending step was modified with an extended (3 hr) T4 PNK incubation prior to adding the T4 polymerase, in the same manner as the UDG protocol. Double-indexing amplification was performed as in (*Wagner et al., 2014*) for 10 cycles each, with 20 µl non-diluted library template input (except L25 and L13 which were used at 0.1x dilution) and included at least one no-template negative control reaction. All reactions were purified to 15 ul EB with the MinElute PCR Purification Kit (Qiagen, Hilden, Germany).

## Enrichment

Two rounds of human mitochondrion targeted enrichment were performed on the non-UDG treated ulna specimen for comparison to the nodule shotgun reads. Prior to enrichment, the 7 Ulna-non-D libraries (L31-L38) were pooled to homogenize input, and 9 ul of this pool was used as input into four enrichment reactions (Ulna-D E07-E10) alongside the extraction blank (E11) and a negative control reaction (enrichment blank). The enrichment reactions were performed as for the human mtDNA enrichments in (*Wagner et al., 2014*), using the same parameters and custom MYbaits baitset (MYcroarray, Ann Arbor, MI) designed from the rCRS sequence (http://www.ncbi.nlm.nih.gov/nucleotide/113200490 NCBI GenBank accession no. J01415.2), but with the following modifications according to updated manufacturer recommendations: post-hybridization bead-library binding was performed rotating at high temp (55°C), Wash Buffer 1 was eliminated, and the post-washed beads were suspended in 20 µl EB and used directly in the post-enrichment amplification. For the adapter-specific blocking oligos, 2 µM of four P5/P7 adapter sequence custom blocking oligos (corresponding to one strand of each molecule) were used for enrichment round 1, and the manufacturer-supplied Block #3 was used for enrichment round 2.

Amplification after each enrichment round was performed as in (*Wagner et al., 2014*), with additional re-amplifications as required due to low output molarities (all amplification reactions were purified over MinElute columns to 15 µl EB). Post-round 1 amplification used 15 µl bead mixture directly as input into each 40 µl reaction (15 cycles). 6.5 µl of this purified reaction was used as input into enrichment round 2 (E17-E21), and to increase molarity prior to sequencing, 6.5 µl was used as a template for subsequent re-amplification reactions (12 cycles). Post-round 2 amplification used 10 µl bead mixture as input into two 40 µl reactions per enrichment (15 cycles), and the supernatants from each amplification (two per sample) were purified together. Prior to sequencing, 14 µl of this purified reaction was used as the template for a subsequent re-amplification reaction (16 cycles).

## Sequencing and read preparation

All relevant ancient samples (nodules, bone and sediment) and their associated extraction blanks were sequenced. Details on the final data set names, their associated samples/libraries/enrichments, raw reads passing filter, and pre-sequencing pooling schemes can be found in *Supplementary file 1K*. Prior to sequencing, all additional indexed libraries (shotgun and enrichment) were quantified via a qPCR assay targeting indexed molecules and pooled according to desired sequencing ratio. All pools except Nod1_all (pool 'K') were size-selected using an electrophoresis gel size selection procedure (retaining molecules ~125/150/150 to 500 bp in length) in order to exclude as much no-insert adaptimer (and other short adapter artifacts) as possible. Pool 'F' also contained 10 additional samples not considered in this paper (pooled at a ratio of '1'). Pools were sequenced across three paired-end runs on the HiSeq 1500, all alongside other, unrelated samples: Pool 'K' (80 bp final read length), Pool 'F' (85 bp final read length), and pools 'G'-'J' (90 bp final read length). On the last run, the enrichment round 1 and 2 samples (pools 'H' and 'J') were separated on two different lanes, since they have the same index sequences.

## Metagenome/Microbiome analyses

For the metagenomic analyses and the ancient pathogen genome reconstructions, all data sets were trimmed of library adapter using cutadapt (*Martin, 2011*) with settings -e 0.16, -O 1, -a AGATCG-GAAGAGC (70) and reads <24 bp were removed retaining read order. To obtain metagenomic profiles from our shotgun data sets we used LMAT version 1.2.3 (*Ames et al., 2013*) to properly identify shotgun reads from the nodules (Nod1-1h-UDG, Nod2-UDG), sediment (Sediment-UDG), ulna (Ulna-UDG), and all metagenomic data sets available from the Human Microbiome Project (HMP, RRID:SCR_012956) database, housed at Lawrence Livermore National Laboratory (June 2015). Reads that could be identified at the sequence level or consistently at the species/strain level from all blank extracts were removed from final files used in the PCA analysis.

The PCAs were performed using prcomp (RRID:SCR_014676) in (*R Core Team, 2015*). The taxa identifications from the HMP were combined according to the origin of the sample. The number of samples combined into each category is indicated in the legend to *Figure 1—figure supplement 8*. A small number ($1 \times 10^{-7}$) was added to those entries with zero reads assigned and then natural logs of the numbers were taken. The PCA was centered and scaled.

## *Staphylococcus saprophyticus* Troy genome reconstruction

Reads were initially processed as described in the previous section. The *S. saprophyticus* Troy draft genome was reconstructed using iterative assembly to span gaps between contigs that were created from assembly to the *S. saprophyticus* reference genome (NC_007350).

Trimmed reads from Nod1_all-UDG were paired-end assembled to the *S. saprophyticus* reference (NC_007350) using BWA (RRID:SCR_010910) with default settings (*Li and Durbin, 2009*), and duplicates were removed using samtools (RRID:SCR_002105) rmdup (*Li et al., 2009*). The resulting assembly was imported into Geneious (v.6.1.6, Biomatters, Ltd, RRID:SCR_010519) and a strict (50%) consensus sequence was generated. From this consensus, 65 large contigs (880 bp – 170,993 bp) that corresponded to regions of average coverage (and that did not span rRNA/tRNA regions) were manually extracted, which represented the non-gap regions of the assembly. As gap regions could represent indel regions (e.g., lateral gene transfer events), rearrangements, or divergence, these contigs were subjected to an iterative assembly process using a set of unmapped reads in order to attempt to span gaps and connect the contigs. The primary set of reads used for iterative assembly was generated by trimming 100 bp from each end of the contigs, assembling all original reads using the above settings to this set of trimmed contigs, and removing these assembled reads from consideration. A subset of the unmapped reads (20–100% as required, depending on assembly success) along with the full set of contigs were then subject to iterative assembly using Geneious (v.6.1.6), seeded with the first or last 50 bp of a contig (settings: maximum gaps per read 10%, maximum gap size 2, word length 20, index word length 14, ignore words repeated >8x, maximum mismatches per read 1%, maximum ambiguity 4, map multiple best matches randomly). All non-rRNA-adjacent gaps were successfully spanned, except for the region that was discovered to belong on the plasmid rather than the chromosome.

## *Gardnerella vaginalis* Troy gene reconstruction

Ancient gene sequences were reconstructed *de novo*, via assembly of a pool of reads that mapped to annotated *G. vaginalis*. First, CDS annotations were extracted from 34 modern *G. vaginalis* assemblies (*Supplementary file 1I*; except for strains 41V and 101) and concatenated into one reference/genome (100 N's between each CDS). Trimmed paired end reads from Nod1_1h-UDG reads were mapped to the concatenated reference. All paired and unpaired reads that mapped were extracted and subjected to *de novo* assembly using Velvet 1.2.1 (*Zerbino et al., 2008*) with settings kmer 23, insert length 51, expected coverage 75, minimum contig length 24, and coverage cutoff auto (parameters for expected coverage were chosen based on previous observation of assembly to strain ATCC 14019). This generated 1207 contigs that were confirmed using blastn (default settings, RRID:SCR_004870) to the nr database (April 2014) to detect any non-*G. vaginalis* sequences or chimeras generated from low level bacterial species also present in the nodules. Twenty contigs were excluded due to the top hit being *S. saprophyticus*, leaving a final set of 1187 *G. vaginalis* contigs (total length 1,435,761 bp). The final set of genomic contigs was annotated using Prokka 1.7 (*Seemann, 2014*) (RRID:SCR_014732) producing 972 genes. Paired-end assembly of Nod1_1h-UDG

reads to the final set of contigs with bowtie2 (*Langmead and Salzberg, 2012*) and removal of duplicates with samtools rmdup (*Li et al., 2009*) consists of 2,034,514 readpairs. Total reads mapping to *G. vaginalis* from paired end-assemblies are listed in *Supplementary file 1L*.

## Bacterial genome coverage and DNA damage estimations

To most conservatively assess the abundance, coverage, and authenticity of our ancient reads, we ran a subset of analyses using slightly more stringent criteria. CASAVA (RRID:SCR_001802) processed reads were trimmed and merged with leeHom (*Renaud et al., 2014*) (RRID:SCR_002710) using its ancient DNA parameter (–ancientdna). We restricted reads from the UDG-treated shotgun nodule libraries (Nod1_1h-UDG and Nod2-UDG) to those of minimum 35 bp length and blasted all reads against the GenBank nucleotide database retaining only the top hit. For all libraries, we additionally mapped to the *S. saprophyticus* ATCC 15305 (NC_007350) and *G. vaginalis* ATCC 14019 (NC_014644) with a customized version of the Burrows-Wheeler Aligner (*Li and Durbin, 2009*) obtained from https://github.com/mpieva/network-aware-bwa) with a maximum edit distance of 0.01 (-n 0.01), allowing for no more than two gaps (-o 2) and with seeding effectively disabled (-l 16500), retaining only those mapped reads which were merged or properly paired [https://github.com/grenaud/libbam/retrieveMapped_single_and_PropertyPair.cpp]. Molecules that were less than 35 bp in length, had a mapping quality score of less than 30, or were marked as duplicates based on both 5' and 3' coordinates were removed [https://bitbucket.org/ustenzel/biohazard.git]. We then pooled all nodule reads DNA originating from the shotgun DNA libraries (Nod1_1h-UDG, Nod1_1h-nonU, Nod2-UDG, Nod2-nonU) and the *S. saprophyticus* and *G. vaginalis* mappings and further removed any duplicated molecules found between libraries (*Supplementary file 1L*). For both mapping assemblies, the average coverage at each reference position was calculated using the bedtools (*Quinlan et al., 2010*) (RRID:SCR_006646) genomecov function and then averaged over 100 bp blocks and visualized with Circos (*Krzywinski, 2009*) (RRID:SCR_011798, *Figure 3—figure supplement 3*, *Figure 4—figure supplement 3*). Fragment length distributions for all pooled libraries and damage plots for the non-UDG treated libraries (Nod1_1h-nonU and Nod2-nonU) were calculated through mapDamage2 (*Jónsson et al., 2013*) (*Figure 3—figure supplement 1*, *Figure 4—figure supplement 1*).

## Human mitochondrial genome analyses

CASAVA processed reads (see directly above), from enriched ulna extractions (Ulna_Enr1-nonU, Ulna_Enr2-nonU), shotgun reads from the UDG treated nodule extraction (Nod1_1h-UDG), and corresponding extraction blanks (EblkD_Enr1-nonU, EblkD_Enr2-nonU and EblkA-UDG) were processed as described above, but mapped to the rCRS mitochondrial genome (NC_012920) (*Andrews et al., 1999*). Consensus sequences were called and contamination was estimated using Schmutzi, which implements iterative probability models to infer the endogenous bases given read length and deamination patterns (*Renaud et al., 2015*). Contamination was estimated at 12% and 13%, respectively, for the round 1 and round 2 enriched ulna libraries. These estimates are consistent with estimates from other aDNA studies (*Posth et al., 2016*). Contamination could not be confidently assessed from the shotgun nodule library as it had been UDG treated, which obfuscates deamination patterns and thereby lessens the differentiation between endogenous and contaminant molecules. mtDNA consensus sequences were uploaded to the HaploGrep webserver [http://haplogrep.uibk.ac.at/] and haplogroups were determined in reference to Phylotree Build 16 (*Kloss-Brandstätter et al., 2011*; *van Oven and Kayser, 2009*) (RRID:SCR_012948) and found to be U3b and U3b3. Haplogroup U3b was assigned to the consensus sequence generated from the first round enrichment of the ulna because there was missing data for polymorphisms diagnostic to the haplogroup U3b3 (*Supplementary file 1E*). All three consensus sequences shared an additional five private polymorphisms not diagnostic to haplogroup U3b3. 137 sequences assigned to haplogroup U3 were collected from all human complete mtDNA genomes in GenBank (18 June 2015), and aligned along with the Troy consensus sequence generated from the nodule shotgun (Nod1_1h-UDG) library with MUSCLE v3.8 (*Edgar, 2004*) (RRID:SCR_011812). It was determined that the best model of nucleotide substitute for this group of 138 sequences was HKY+I+Γ using the program jModelTest2 (*Darriba et al., 2012*) and the built-in Akaike Information Criterion (*Akaike, 1974*). A Bayesian Maximum Clade Credibility tree was calculated using BEAST v1.8 (*Drummond et al., 2012*) (RRID:SCR_

010228) and TreeAnnotator (*Drummond et al., 2007*) with the nucleotide data partitioned between coding and non-coding and a strict molecular clock with evolutionary rates of $1.708 \times 10^{-8}$ and $9.883 \times 10^{-8}$ nucleotide substitutions/site/year following Soares et al. (*Soares et al., 2009*) (*Figure 1—figure supplement 6*). Damage patterns and fragment length distribution of ancient DNA mapped to mitochondrial genome can be found in *Figure 1—figure supplements 3–4*.

## Human nuclear genome analyses

Reads from four shotgun libraries (three from the nodules, Nod1_1 hr_UDG, Nod2-UDG, Nod2-nonU; one from the ulna, Ulna-UDG) were mapped and processed as described for the mitochondrial genome above. Additionally, the reads originating from the four nodule libraries were pooled together for comparison ('Nodule Pooled'). We mapped the libraries to a hard-masked hg38 reference genome downloaded from the UCSC genome browser [http://hgdownload.soe.ucsc.edu/goldenPath/hg38/bigZips/] and recorded the number of reads mapping to chrX, chrY, mitochondrial genome and all autosomes. We first filtered for mapped merged or properly paired reads with a minimum length of 35 bp and a minimum mapping quality filter of 30. Percent coverage was calculated by tallying the number of positions covered by at least one read and dividing by the total genome length with masked regions subtracted. We calculated the coverage depth by summing coverage of all positions and dividing the total by this same masked genome length (*Supplementary file 1G*).

## Extraction and sequencing of modern *S. saprophyticus*

Fourteen new *S. saprophyticus* isolates (North America: eight human, one bovine, and one canine; Australia: two human; Japan: two human) were sequenced for this study to provide a broader comparative genomic data set (*Supplementary file 1H*).

### Extraction and sequencing of North American *S. saprophyticus*

DNA was extracted at the University of Wisconsin-Madison. Isolates were inoculated in TSB and grown overnight at 37°C in a shaking incubator. Cultures were pelleted, resuspended in 140 µL TE, and incubated overnight with 50 units of mutanolysin. DNA was extracted using the MasterPure Gram Positive DNA Purification Kit (EpiCentre). Extracts were prepared for sequencing with the Illumina Nextera XT library preparation kit and sequenced in two different batches: samples 13–31 were pooled in equal ratio with 24 additional unrelated samples and sequenced on a MiSeq platform using a 2×250 kit; samples K, X, and M were sequenced at a desired ratio alongside pools 'G'-'J' (described above) as part of the HiSeq 1500 paired-end run (85 bp read length). Average insert sizes (estimated from Agilent Bioanalyzer analysis) are 849 bp (13–31, pooled average), 775 bp for M, 716 bp for X, and 769 bp for K.

### Extraction and sequencing of Australian *S. saprophyticus*

DNA extractions were performed as described above. DNA was submitted to the University of Wisconsin-Madison Biotechnology Center. DNA concentration was verified using the Qubit dsDNA HS Assay Kit (Life Technologies, Grand Island, NY). Samples were prepared according the TruSeq Nano DNA LT Library Prep Kit (Illumina Inc., San Diego, California, USA) with minor modifications. A maximum of 200 ng of each sample was sheared using a Covaris M220 Ultrasonicator (Covaris Inc, Woburn, MA, USA). Sheared samples were size selected for an average insert size of 550 bp using Spri bead based size exclusion. Quality and quantity of the finished libraries were assessed using an Agilent DNA High Sensitivity chip (Agilent Technologies, Santa Clara, CA) and Qubit dsDNA HS Assay Kit, respectively. Libraries were standardized to 2 µM. Paired-end, 150 bp sequencing was performed using v2 SBS chemistry on an Illumina MiSeq sequencer. Images were analyzed using the Illumina Pipeline, version 1.8.2.

### Extraction and sequencing of Japanese *S. saprophyticus*

*S. saprophyticus* cells were lysed by achromopeptidase (WAKO, Kyoto, Japan), and the genomic DNA was prepared with conventional phenol/chloroform extraction and ethanol precipitation, followed by further purification with QIAGEN genome DNA preparation kit. Library preparation was completed using the Nextera XT DNA Sample Prep Kit (Illumina, San Diego, CA, USA), followed by insert size selection using 1% TAE agarose electrophoresis to obtain an insert of approximately 400

bp. Sequencing was performed on NextSeq 500 (Illumina, San Diego, CA, USA) using the NextSeq 500 v1 kit (300 cycle) with paired-end 150 bp sequencing.

## Reference-guided assembly and alignment of *S. saprophyticus*

Reads for the new modern samples were processed with reference-guided assembly via a pipeline [https://github.com/tracysmith/RGAPepPipe]. For reference guided assembly, read quality was assessed and trimmed with TrimGalore! v 0.4.0 [www.bioinformatics.babraham.ac.uk/projects/trim_galore], a wrapper script for FastQC [www.bioinformatics.babraham.ac.uk/projects/fastqc, RRID: SCR_005539] and cutadapt (*Martin, 2011*) (RRID:SCR_011841). Reads were mapped to the ATCC 15305 reference genome using BWA-MEM v 0.7.12 (*Li, 2013*) (RRID:SCR_010910) and bam files sorted using Samtools v 1.2 (*Li et al., 2009*) (RRID:SCR_002105). Read group information was edited and duplicates removed using Picard v 1.138 [picard.sourceforge.net, RRID:SCR_006525]. Reads were locally realigned using GATK v 2.8.1 (*DePristo et al., 2011*) (RRID:SCR_001876). Variants were called using Pilon v 1.16 (*Walker et al., 2014*) (RRID:SCR_014731) with a minimum read depth of 10, minimum mapping quality of 40 and minimum base quality of 20. Whole genome alignment of the Troy strain and *de novo* assemblies to ATCC 15305 was performed using Mugsy 2.3 (*Angiuoli et al., 2011*) (RRID:SCR_001414).

## *De novo* assembly and annotation of *S. saprophyticus*

The draft genome sequences of Japanese isolates were obtained by *de novo* assembly using CLC genome workbench v8.02 (RRID:SCR_011853). For North American and Australian *S. saprophyticus* genomes, *de novo* assembly was performed using the iMetAMOS pipeline (*Koren et al., 2014*; *Treangen et al., 2013*) (RRID:SCR_011914). We compared assemblies from SPAdes (*Bankevich et al., 2012*), MaSurCA (*Zimin et al., 2013*), and Velvet (*Zerbino et al., 2008*). KmerGenie (*Chikhi and Medvedev, 2014*) was used to select kmer sizes for assembly. iMetAMOS uses FastQC [www.bioinformatics.babraham.ac.uk/projects/fastqc] to check read data quality. Assemblies were evaluated using QUAST (*Gurevich et al., 2013*), REAPR (*Hunt et al., 2013*), LAP (*Ghodsi et al., 2013*), ALE (*Clark et al., 2013*), FreeBayes (*Garrison and Marth, 2012*), and CGAL (*Rahman et al., 2013*). Additionally, Kraken (*Wood et al., 2014*) was run to detect potential contamination in sequence data. The SPAdes assembly was identified as best for isolates 13, 16, 19, 41, 42, 43, K, M, X, and 129. The MaSurCA assembly was identified as best for isolates 14, 15, 18, and 31. Genomes were annotated using Prokka 1.7 (*Seemann, 2014*) (RRID:SCR_014732). OrthoMCL v2.0.9 (*Li et al., 2003*) (RRID:SCR_007839) was used to find orthologous genes in these genomes.

## Core genome alignment of *G. vaginalis*

Genomes were annotated with Prokka 1.7 (*Seemann, 2014*) (RRID:SCR_014732). OrthoMCL v2.0.9 (RRID:SCR_007839) grouped genes into orthologous groups (*Li et al., 2003*). Genes were filtered to include only genes present in one copy in every genome. Individual genes ($n$ = 537) were aligned with TranslatorX (RRID:SCR_014733) and MAFFT v7.130b (*Abascal et al., 2010*; *Katoh and Standley, 2014*) (RRID:SCR_011811) and concatenated [https://github.com/tatumdmortimer/core-genome-alignment].

## Phylogenetic analyses

Maximum likelihood phylogenetic trees were inferred using RAxML 8.0.6 (*Stamatakis, 2014*) (RRID: SCR_006086). Bootstrap replicates (number determined by autoMR convergence criteria) were applied to the tree with the highest likelihood of twenty using the GTRGAMMA substitution model. We used SplitsTree4 (*Huson and Bryant, 2006*) (RRID:SCR_014734) to create networks of pSST1, the chromosome in *S. saprophyticus*, and the core genome of *G. vaginalis.*

## Recombination

Recombination in a whole genome alignment of *S. saprophyticus* isolates and a core genome alignment of modern *G. vaginalis* isolates was assessed using BratNextGen (*Marttinen, 2012*). For both *S. saprophyticus* and *G. vaginalis* analyses, one hundred permutations were performed to calculate the significance (p<0.05) of recombinant fragments (plots created with Circos [*Krzywinski, 2009*]). Recombination was also measured for the pSST1 plasmid alignment using Phi (*Bruen et al., 2006*),

Max $\chi^2$ (*Smith, 1992*), and NSS (*Jakobsen and Easteal, 1996*) implemented in PhiPack; results of these tests were all significant with p-values of $5\times10^{-15}$, 0, and 0, respectively.

## Variant annotation

We used SNP-sites (*Page et al., 2016*) to convert the alignment of *S. saprophyticus* isolates to a multi-sample VCF. SnpEff (*Cingolani, 2012*) (RRID:SCR_005191) was used to annotate variants in this VCF as synonymous, non-synonymous, or intergenic.

## Analysis of temporal structure

To determine whether there was sufficient temporal structure in the *S. saprophyticus* phylogeny to estimate evolutionary rates, we performed a regression of root-to-tip genetic distances against year of sampling using TempEst v 1.4 (*Rambaut et al., 2016*). We also attempted to estimate evolutionary rates using BEAST v1.8 (*Drummond et al., 2012*) (RRID:SCR_010228). Results of both these analyses (Appendix) revealed a lack of temporal structure such that rate (and date) estimates are unreliable.

## Analysis of population structure

We used a Bayesian tree sampling method implemented in BaTS (vBETA2) (*Parker et al., 2008*) to determine the significance of phylogenetic clustering and population structure in our *S. saprophyticus* data. A distribution of phylogenies was generated using BEAST v1.8 (*Drummond et al., 2012*) (RRID:SCR_010228) with GTR+Γ substitution model, a strict molecular clock, and a constant population size. Markov chains were run in duplicate for 10 million generations each with sampling every 1000 generations, and the first 1 million generations were discarded as burn-in.

## Acknowledgements

We thank the Troia Project archaeological team, especially Gebhard Bieg, for providing insights and images. We thank Dr. Ernst Pernicka, who authorized release of the material for study, and the General Directorate of Museums and Monuments in the Ministry of Culture and Tourism of the Republic of Turkey for permissions to conduct research in Turkey and take samples. We also thank Kurt Reed, Pam Ruegg, Scott Weese, Vitali Sintchenko, and Amanda Harrington for providing *S. saprophyticus* isolates. The authors would like to acknowledge the efforts of Tracy Smith and Mary O'Neill at University of Wisconsin-Madison for growth, extraction, and library preparation of *S. saprophyticus* isolates. We also thank the University of Wisconsin Biotechnology Center DNA Sequencing Facility for providing library preparation and sequencing facilities and services. Victoria Jarvis of the Brockhouse Institute for Materials Research and Henry Schwarz of the Department of Geology (McMaster University) helped with XRD, and Marcia Reid of the Electron Microscopy Facility (McMaster University) provided assistance with TEM/SEM-EDS analyses. We acknowledge helpful feedback from the members of the McMaster Ancient DNA Centre and the Pepperell Lab in our ongoing studies of ancient and extant pathogen genomes.

## Additional information

### Funding

| Funder | Grant reference number | Author |
| --- | --- | --- |
| Canada Research Chairs | | Hendrik N Poinar |
| Natural Sciences and Engineering Research Council of Canada | | Hendrik N Poinar |
| National Institutes of Health | National Research Service Award, T32 GM07215 | Tatum D Mortimer |
| National Science Foundation | Graduate Research Fellowship Program, DGE-1256259 | Tatum D Mortimer |

| National Institutes of Health | R01AI113287 | Caitlin S Pepperell |
| --- | --- | --- |
| McMaster University | Michael G. DeGroote Institute for Infectious Disease Research (IIDR) | Hendrik N Poinar |
| University of Wisconsin-Madison | Graduate School | William Aylward Caitlin S Pepperell |
| Wisconsin Alumni Research Foundation | | William Aylward Caitlin S. Pepperell |

The funders had no role in study design, data collection and interpretation, or the decision to submit the work for publication.

## Author contributions

AMD, HNP, Conception and design, Acquisition of data, Analysis and interpretation of data, Drafting or revising the article; TDM, Conception and design, Analysis and interpretation of data, Drafting or revising the article; AK, GBG, ATD, SNG, JEA, DEGB, GF, ECH, Analysis and interpretation of data, Drafting or revising the article; HK, WA, Acquisition of data, Drafting or revising the article, Contributed unpublished essential data or reagents; JME, JS, AMK, GW, Acquisition of data, Analysis and interpretation of data, Drafting or revising the article; MKuc, MKur, KK, Acquisition of data, Drafting or revising the article; CSP, Conception and design, Acquisition of data, Analysis and interpretation of data, Drafting or revising the article, Contributed unpublished essential data or reagents

## Author ORCIDs

Tatum D Mortimer, http://orcid.org/0000-0001-6255-690X
Edward C Holmes, http://orcid.org/0000-0001-9596-3552
Caitlin S Pepperell, http://orcid.org/0000-0002-6324-1333

## Ethics

Human subjects: Ethics approval for the study of the remains of the individual excavated in 2005 from grave 14 (Troy project, University of Tubingen, bone-sample x24.177) in quadrat x24 at Troy was obtained from Hamilton Health Sciences and McMaster University (REB# 13-146-T). Samples of extant bacteria were provided to investigators without patient identifiers or protected health information; the members of the study team did not have access to any identifiers or protected health information associated with the bacterial isolates.

## Additional files

### Supplementary files

• Supplementary file 1. (A) Troy sample details (B) SEM-EDS results from nodule. For each replicate, upper value is weight %, lower value is atomic %. (C) Common chemical constituents of renal and bladder calculi (kidney and bladder stones) and Troy nodules. + - presence, ND- not detected, Unk-unknown, RF- Relative Frequency in modern populations (C.Y.C Pak (ed.) *Pak [1987]*, Martinus Nijhoff Publishing, Boston). (D) Relative frequency of admixed calculi (kidney and bladder stones) in modern (grey shading) populations and archaeological findings (modified from *Pak, 1987*). Acronyms correspond to Table C. References cited herein. *1 Main constituent of bladder stone listed as ammonium acid urate and oxalate. *2 Majority of calculus determined to be calcium carbonate (calcite). Where not specifically given, relative proportions of all mineral components were estimated. *3 Minor constituents of calcium carbonate. *4 No other (minor) elements were provided thus composition was assumed to be 100%. ND – not determined/detected. (E) Mitochondrial results summary; unique reads mapped to the revised Cambridge Reference Sequence with minimum length of 35 bp and minimum mapping quality of 30. (F) Sex identification of all libraries after *Skoglund et al. (2013)*. Libraries were mapped to a hard masked version of hg38 and restricted to reads of minimum length 35 bp and minimum mapping quality of 30. (G) Summary of unique shotgun reads (from nodules, ulna, sediment and associated blanks) of minimum length 35 bp and minimum mapping

quality of 30, mapping to chromosome X, Y, autosomes and mitochondrion of the hard masked hg38. Nodule pooled =Nod2-UDG + Nod1. (H) *Staphylococcus saprophyticus* modern strains. (I) *Gardnerella vaginalis* modern strains. * F = full, S = scaffold; ** BWH = Brigham and Women's Hospital, HMP = Human Microbiome Project, MWH = Magee-Womens Hospital, VCU = VCU Women's Health Clinic (J) Troy DNA extraction details. *Demin. = demineralization; digest = digestion buffer (K) HTS data sets. (L) Summary of unique shotgun reads (from nodules, ulna, sediment and associated blanks) of minimum length 35 bp and minimum mapping quality of 30, mapping to *Staphylococcus saprophyticus* and *Gardnerella vaginalis*. Nodule pooled =Nod2-UDG + Nod1_1h-nonU + Nod2-nonU + Nod1_1h-UDG). (M) Kinetic analysis of ancient PC1 $\beta$-lactamase (N) *S. saprophyticus* Clade P specific genes. Locus tags refer to annotation of ATCC 15305 available at NCBI. (O) Non-synonymous variants shared between *S. saprophyticus* Clade P and isolate 55. REF is reference allele in ATCC15305, Clade P and isolate 55. ALT is allele in the remaining Clade E isolates. Locus tags refer to annotation of ATCC 15305 available at NCBI.

### Major datasets
The following datasets were generated:

| Author(s) | Year | Dataset title | Dataset URL | Database, license, and accessibility information |
|---|---|---|---|---|
| Henrike Kiesewetter, William Aylward, Caitlin S Pepperell, Tatum Mortimer, Alison M. Devault, Ana T Duggan, Hendrik N Poinar, Jacob M Enk, Melanie Kuch | 2016 | Ancient DNA sequencing of calcified nodules from Late Byzantine Troy | https://www.ncbi.nlm.nih.gov/bioproject/PRJNA352376 | Publicly available at the NCBI (accession no: PRJNA352376) |
| Caitlin S Pepperell, Tatum Mortimer, Makoto Kuroda, Kengo Kato | 2016 | Whole genome sequencing of human and animal associated *Staphylococcus saprophyticus* | https://www.ncbi.nlm.nih.gov/bioproject/PRJNA352403 | Publicly available at the NCBI (accession no: PRJNA352403) |

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

## Ancient data generation and analyses

### Archaeological context

Of approximately 200 total burials found in controlled excavations at Troy between 1988 and 2012, 102 were found in three separate Late Byzantine cemeteries around the periphery of the ancient city (*Kiesewetter, 2014*; *Jablonka, 2006*). One of these (*Figure 1—figure supplement 1*), in quadrat x24, excavated in 2005, yielded 30 individuals from sixteen graves, including the woman described in the main text (Grave 14, context x24.177). The cemetery was dated to the early thirteenth century A.D., based on stratigraphy, topographical situation and relationship to other monuments at Troy, grave construction techniques and mortuary practices, pottery found in and around the graves, and comparison to other Late Byzantine graveyards at Troy (*Kiesewetter, 2014*; *Jablonka, 2006*).

### Description of osteological materials *Grave 14, Ind. I, Beh.177*

In grave 14 (Troy, area x24) bones from one single individual (Beh.177) were unearthed. According to the overall appearance and morphology, the bones were fairly well preserved with minor signs of degradation. The skeleton was found in situ when excavated. The deceased was lying in a supine position with crossed arms, skull in the northwest facing southeast (*Figure 1A*). From the skull only the calvaria (cranial vault) was well preserved, the facial part including the maxilla was missing, and from the mandible only the left portion was present. No teeth were found, which is also due to ante-mortem tooth loss observed on the mandible. Both scapulae were present but fragmented. The right clavicle was missing; the left clavicle was preserved, but distal and proximal ends were missing. All long bones were represented; the proximal half of the right humerus was missing, right radius and ulna were complete, the left arm was almost complete, both femora and tibiae were almost complete, and from the fibulae only fragments remained (*Figure 1—figure supplement 9*). Ribs and vertebrae were fragmented, and some thoracic vertebrae were missing. The left pelvis was well preserved, while the right pelvis was very brittle and fragmentary. Metacarpals and metatarsals were almost completely preserved while other hand and foot bones were present but fragmented.

The recording of sex differences and age changes as well as the sex and age estimation followed the scheme recommended by the *Standards for Data Collection from Human Skeletal Remains* (*Buikstra and Ubelaker, 1994*).

### Sex estimation

### Sex estimation using the pelvic bones

The ventral arc and the shape of the greater sciatic notch of the pelvis demonstrated female features as did the subpubic concavity, the ischiopubic ramus ridge, the preauricular sulcus, and the composite arch of the pelvis (*Buikstra and Ubelaker, 1994*; *Bruzek, 2002*).

### Sex estimation according to cranial morphology

The skull revealed slight supra-orbital ridges, sharp orbital borders, minimal expression of the rugosity of the nuchal crest, and a small mastoid process. All these features indicate a female sex of the skeleton. Only the glabella demonstrated an ambiguous feature (*Buikstra and Ubelaker, 1994*).

## Age estimation

### Age estimation for adult crania

Age estimation was made using the degree of cranial suture closure (*Meindl and Lovejoy, 1985*). The spheno-occipital synchondrosis was completely fused. The sagittal suture at the part of the obelion demonstrated significant closure, other cranial sutures showed no or minimal closure suggesting that the skeleton belonged to a 'young adult' no older than 35 years.

### Age estimation of the postcranial skeleton

Age estimation was made using the degree of epiphyseal fusion (*Scheuer, 2000*). The epiphyses of all long bones were completely fused indicating an adult age over 20 years. The complete epiphyseal fusion of all parts of the spinal column suggests an age over 25 years (*Scheuer, 2000*).

The cranial and postcranial examination together indicate that the skeletal remains are most likely from a female individual whose age-at-death would have been between 25 and 35 years. The appearance of the surfaces of the joints and the pubic symphyseal face support this age estimation.

### Pathological findings

The skull showed signs of slight orbital pitting (*cribra orbitalia*). Only the left mandible was preserved and all teeth were lost ante-mortem. Lumbar vertebrae 3 to 5 show slight to medium vertebral degeneration with osteophytic lipping of approximately 2–5 cm. The left ulna showed signs of a well-healed fracture of the styloid process. The left ulna and radius also revealed signs of severe osteomyelitis.

| | |
|---|---|
| Estimated height | 159 cm |
| Max femur length | 397 cm |
| Max tibia length | ca. 315 cm |
| Max ulna length | 231 cm |

### Radiocarbon dating

High molecular weight (>30 kDa) ultrafiltered collagen extracted from the ulna (sample no x24.177) following (*Beaumont et al., 2010*) and graphitized and measured using standard AMS techniques returned a $^{14}$C age of 860 ± 20 BP (UCIAMS# 131407). The corresponding calibrated date is 1154–1224 AD (0.98 probability distribution at two sigma) which is in agreement with the archaeologically established date for the quadrat x24 Troy burials in the 13th century AD (*Kiesewetter, 2014*; *Jablonka, 2006*). The collagen yield of 4.5% and the carbon and nitrogen content (16.0% N, 46.1% C, atomic C/N ratio 3.3) are typical of results from well preserved bone, and the stable isotope values ($^{13}$C = −19.6‰, $^{15}$N = + 8.2‰) suggest an individual whose dietary protein was largely derived from terrestrial plant sources. The diversity of $^{14}$C dates (460–1130 yrs BP, *Supplementary file 1A*) from the carbonate and organic fractions of the nodules (for example, the extremely young date on the 10% leached carbonate sample) is suggestive of possible ground water leaching and exchange. We made an additional attempt to date organic residues from demineralized nodule one, using base treatment (0.1N NaOH) to remove possible soil carbon contamination. This resulted in a very old date of 4680 yr BP, possibly due to the presence of residual refractory soil carbon (humin). Given these discrepant ages, the results for the nodules must be viewed with caution, in contrast to that of the bone.

## Bacterial mineralization and biofilm formation

Soft tissue mineralization, enabling the long-term morphological preservation of cellular details, is not uncommon in the fossil record and often involves the replication of organic constituents in calcium phosphate (apatite) (*Briggs and Kear, 1993*; *Briggs, 2003a, 2003b*).

This process is controlled by the concentration of available phosphate and by pH, which must be at levels that inhibit the formation of calcium carbonate (*Briggs and Wilby, 1996*; *Briggs and Kear, 1994*). The role of commensal and environmental bacteria in the decomposition of soft tissues is becoming increasingly clear (*Briggs, 2003a*; *Metcalf et al., 2016*). Despite their ubiquity, however, bacteria themselves decay rapidly and are only occasionally mineralized in fossils, where they may be preserved in association with soft tissues (within-host environments) (*Briggs et al., 1997, 2005*). Melanosomes, which are commonly fossilized in association with colored structures such as feathers (*Vinther et al., 2008*), were originally mistaken for fossilized bacteria (*Wuttke, 1983*), an interpretation that persists in some cases (*Moyer et al., 2014*), but melanin, in contrast to bacteria, is very resistant to degradation (*Vinther, 2016*). Soft tissue fossilization relies on microbial activity to release mineral-forming ions and establish geochemical gradients, and microbial films (biofilms) may control diffusion (*Briggs, 2003a*). Biofilms, which are composed of spatially organized bacterial cells attached to a common surface and embedded within a matrix of extracellular polymeric substances (EPS - a glycocalyx secreted by the cells), rarely preserve cellular details, but individual microbial morphologies, such as coccoids, may become mineralized (*Briggs, 2003b*; *Briggs et al., 1997*).

Scanning Electron Micrograph (SEM) images of the Troy nodules taken at various magnifications (*Figure 2*) appear to show bacterial cells (and likely biofilms) within a labyrinthine structure reminiscent of the complex arrangement of overlapping villi visualized in SEM of the placenta (*Hafez and Kenemans, 2012*). We interpret the coccoid structures evident in the images (*Figure 2*, red arrows) as the superficially mineralized remains of bacteria. The rounded shape and clustering (without chain formation) are typical of *Staphylococci*, which measure between 0.5 and 1.0 microns in diameter. The size of the cocci in the electron micrographs (*Figure 2*) vary between 0.5 to 3.5 microns; the range in size may represent cells with varying degrees of biomineralization (i.e. precipitation of apatite and carbonate on the cell wall) thereby increasing their overall dimensions. The structures in *Figure 2* (yellow arrows) have an appearance typical of EPS, suggesting that the *Staphylococci* were organized into a biofilm during active infection. Given the likely origin of the nodules studied here (see below), the presence of biofilms is not surprising. Biofilms are commonly associated with chronic infections and have been found in many tissues, including within the amniotic cavity (*Romero et al., 2008*). Larger irregular structures (*Figure 2—figure supplement 2*, blue arrows) more than five microns in size, are likely to represent neutrophils, which would be expected at the site of an active infection. The nodule appears to capture a 'Kodak' moment, with the cellular combatants of severe infection frozen in time by replication in minerals.

## Calcified placental abscesses vs urinary calculi

The XRD analysis of 'Troy' nodule one (*Figure 2—figure supplement 1*) shows it to be predominantly composed of two phosphate phases, hydroxlyapatite (bioapatite - $Ca_5(PO_4)_3(OH)$) and whitlockite ($Ca_3(PO_4)_2$) as well as small amounts of calcium carbonate (calcite- $CaCO_3$), both of which have been found in pathological calcified concretions (*Lagier and Baud, 2003*). Early morphological assessment of the nodules (multistratified concentric layering, overall shape and size) suggested possible renal or bladder calculi. Urinary calculi (kidney/bladder stones) are mineral aggregates formed under a variety of different physiological conditions (*Balaji and Menon, 1997*) and composed predominantly of whewellite, weddellite (two forms of calcium oxalate ($CaC_2O_4H_2O$)), struvite (magnesium

ammonium phosphate - $Mg(NH_4)(PO_4)6H_2O$) and hydroxylapatite (bioapatite - $Ca_5(PO_4)_3(OH)$) with other, minor mineral components (*Supplementary file 1C,D*) (*Pak, 1987*). *Supplementary file 1D* compares the relative frequency of these mineral components (individually and in combination) found within kidney and bladder stones in modern populations sampled from the US, UK and Germany, as well as material identified as possible stones from archaeological excavations. Most modern and ancient kidney stones (*Supplementary file 1D*) are composed predominantly of calcium oxalate, struvite and apatite, and in a few cases small amounts of ammonium acid urate and calcium carbonate (*Steinbock, 1989*; *Giuffra et al., 2008*; *D'alessio, 2005*; *Giuffra, 2011*). Ancient bladder stones have also been shown to be comprised of calcium oxalate, struvite and calcium carbonate. Interestingly a few large stones identified in archaeological material (*Brothwell and Sandison, 1967*; *Piperno, 1976*; *Szalai and Jávor, 1987*; *Hawass and Brock, 2003*; *Anderson, 2003*; *Özdemir et al., 2015*) have a predominantly apatite or carbonate-phosphate signature similar to the Troy nodules (*Supplementary file 1D*). While most earlier works (1967–1987) lack specific elemental details, precluding a more thorough comparison, the recent analysis of an ancient bladder stone from Oluz Höyük showed it to be composed almost exclusively of calcium phosphate (*Özdemir et al., 2015*). However, without further molecular/metagenomic characterization, it is difficult to pinpoint the tissue of origin of these ancient stones, or to compare them with the nodules studied herein. Notwithstanding the small sample size, the absence of calcium oxalate and struvite – as seen in XRD and SEM-EDS analyses of the Troy nodules (average 42.3% oxygen, 42.1% calcium, 7.53% phosphate, and 3.18% magnesium, by weight; *Supplementary file 1B*) – argues against a diagnosis of kidney or bladder stones.

The retrieval of DNA stemming predominantly from *S. saprophyticus* and *G. vaginalis* within the nodules suggests a urogenital origin for the calculi, although these bacteria do not pinpoint the tissue of origin (i.e. tubo-ovarian abscess, endometrium, cervix, vagina, or renal tract). While both *S. saprophyticus* and *G. vaginalis* can be found in the vaginal flora (*Higashide et al., 2008*; *Harwich et al., 2010*), *S. saprophyticus* is primarily associated with urinary disease (UTI) and both genera may occur in the female urinary microbiome (*Lewis et al., 2013*; *Fouts et al., 2012*). *S. saprophyticus* infection has been directly associated with urinary calculi in humans (*Fowler, 1985*) and rats (*Gatermann et al., 1989*) although these calculi are comprised predominantly (~52%) of struvite (magnesium ammonium phosphate) and apatite (~38%) in humans, and almost exclusively struvite in rats. Rates of stone disease are typically higher in men than in women (2.7:1, 2006) except in cases of infection-associated calculi, where they are higher in women (3:1) (although again these stones are composed almost entirely of struvite, magnesium ammonium urate, newberyite and ammonium hydrogen urate and not apatite) (*Knoll et al., 2011*).

Dystrophic or ectopic calcifications of various tissues (resulting in calculi, similar in shape and size to urinary stones) can occur as a result of infectious disease, injury or aging (*Giachelli, 1999*; *Ronchetti et al., 2013*). Calcification of the placenta in particular is common, and is associated with both normal pregnancies and a range of pathological conditions. Placental calcifications occur in most pregnancies after week 33 (>50%) and are universal in post-date pregnancies (*Tindall and Scott, 1965*; *Spirt et al., 1982*). Chorioamnionitis, a potentially fatal infection in which urogenital bacteria gain access to the amniotic membranes and placenta (*Redline and Frcpa, 2007*), can also result in marked tissue calcification. When chorioamnionitis progresses to involve the umbilical vessels (funisitis), for example, concentric rings of calcification are commonly visible to the naked eye and umbilical cord mineralization may be so pronounced that the cord cannot be clamped (*Kraus et al., 2004*; *Benirschke et al., 2006*). Based on the propensity of placental tissues to undergo dystrophic calcification, the elemental composition of the Troy nodules, their appearance under SEM, the urogenital bacteria recovered, as well as the age and sex of the decedent, chorioamnionitis is a probable underlying diagnosis. Reproductive-associated infections can cause premature labor and maternal and fetal sepsis (*Redline and Frcpa, 2007*). These and other obstetric complications were

undoubtedly major factors in the mortality of women throughout history (*Sayer and Dickinson, 2013*), as they remain today (*WHO et al., 2014*). Archaeological palaeodemographic findings from Late Byzantine Troy reveal low overall life expectancies – 32 for females, 39 for males – and a higher mortality rate for females of reproductive age (*Kiesewetter, 2014*), emphasizing the importance of pregnancy complications as a cause of death among women in this setting.

Urolithiasis is an alternative diagnosis for the nodules. We favor chorioamnionitis for the following reasons. First, while urolithiasis can develop at any age, young women have the lowest rates of disease across a range of settings (*Romero et al., 2010*). By contrast, paleodemographic data implicate pregnancy complications as the leading cause of death among women in the Late Byzantine world (*Bourbou, 2010*). Urinary stones commonly harbor viable bacteria (*Tavichakorntrakool et al., 2012*), and urinary tract infection with *S. saprophyticus* and other urease-producing bacteria may lead to stone formation (*Fowler and Jackson, 1985*). Elemental analyses of the nodules were not, however, typical of either infection-related (i.e. struvite) or other types of urinary stone. The abundance of human DNA in the nodules also argues for their origin in human tissue, as opposed to inorganic stone material with entrapped human cells. Lastly, a diagnosis of urinary stones does not explain the endogenous Y chromosome in the nodules.

## Authenticity of ancient chrY reads and attempts to sex the remains

To ensure to the greatest extent possible that reads mapping to Y chromosome were endogenous, we looked at the FLD mean and median of the hard masked, pooled, mapped fragments as well as their damage profiles (from all four nodule libraries). The median for reads (>35 bp, $Q_m$ = 30) mapping to the X, Y and autosomes were 45, 44 and 45 bp respectively and thus nearly identical suggesting that they stem from the same pool of damaged human endogenous DNAs found within the abscess. The damage profiles for these same reads, restricted to the two non-UDG treated shotgun libraries) clearly shows damage in those mapping to the X and the autosomes. While there is more noise in the Y, due to overall fewer reads present, they still have a decided increase in C to T and G to A transitions at terminal positions, again suggestive of their authenticity (*Figure 1—figure supplement 5*). We used a Fisher's exact test to assess whether the number of unique reads mapping to the Y chromosome was significantly greater in the nodule than the ulna or sediment samples, despite unequal sequencing depths (893/33262468 nodule vs 3/4558127 bone). We found the nodule to have significantly more Y-mapped reads than either the ulna and sediment samples (p<2.2E-16 for both tests). There was no significant difference in the proportion of Y-mapped reads between the ulna and sediment samples. We attempted to sex the remains (i.e calcified nodules, ulna and their associated blanks as well as the surrounding sediments and blanks) using the method developed and published by *Skoglund et al. (2013)*. Our input consisted of all reads >35 bp, mapping to the hard masked version of hg38 with map quality >30. As can be seen (*Supplementary file 1F*) all libraries derived from nodule DNA extractions are consistently assigned as female (XX). It is worth noting that in addition to this assignment (XX), all libraries also show the consistent presence of Y reads. The sediment has no assigned sex and the ulna is 'consistent with XX' but read numbers are low (n = 22). Importantly, blanks, also with low read counts (n = 1–35) are labelled as 'consistent with XX' with 95% confidence intervals (at those levels) of 0. Considering (a) that the skeleton is biologically female and of child-bearing age, (b) that the nodule and ulna share the same mtDNA haplotype, and (c) the relative abundance of X chromosome reads in both nodules, we interpret the significantly higher Y-mapped reads in the nodule over the ulna as evidence to support that the human DNA recovered from the nodule originated from that of two individuals, a female and her male fetus.

### *Staphylococcus saprophyticus* Troy genome

#### Resolving duplicated/dual-motif regions

SSP0354-0356. There is an apparent divergent paralogous/duplicated region of NC_007350 genes SSP0354 (ispD; 2-C-methyl-D-erythritol 4-phosphate cytidylyltransferase), SSP0355 (zinc-binding dehydrogenase) and SSP0356 (glycosyl glycerophosphate transferase involved in teichoic acid biosynthesis) inside of the novel GI. The duplicated region precluded straightforward iterative or *de novo* assembly due to challenges in resolving which version was found in the same location as the reference and which version was found in the novel region. The challenges arose from the partially conserved, partially highly divergent nature of the duplicated region (which is problematic for both reference-guided and *de novo* assembly) as well as the short reads precluding the spanning of SNPs (haplotypes) by long paired-end reads. Proper gene sequences were established via a combination of (a) manually working inwards from the shared flanking regions with liberal assembly to manually determine correct sequence and (b) iterative assembly to *de novo* reconstruct highly divergent sections. For the majority of positions, the ancient versions of SSP0354-56 do match the reference genome and the novel copies are the 'unmatching' positions; however, in several instances, the reads support that the 'reference matching' positions are in fact on the paralogous copies instead of the homologous ancient SSP0354-56 genes. SSP0117, SSP0118, and SSP0788. As for SSP0354-0356 and their paralogs discussed above, the sequences of the Troy versions of the SSP0117, 0118, and 0788 paralogous hypothetical proteins were obtained through a combination of manually working inwards from the shared flanking regions with liberal assembly to manually determine correct sequence and iterative assembly to *de novo* reconstruct highly divergent sections.

#### *S. saprophyticus* Troy genome verification

All potential reads were assembled to a penultimate draft version of the genome (a combined product of reference-guided assembly consensus contigs, iterative *de novo* assembly, and manual curation as noted). Ambiguous positions (from the contig and iterative assembly consensus sequences) or positions heterogeneous in the reads (>40%) were resolved with the reference base (if that motif was present in the reads >40%). All Nod1_1h-UDG reads were assembled to the final genome draft using paired-end bowtie2 assembly (**Langmead and Salzberg, 2012**) with default settings and samtools rmdup (**Li et al., 2009**), and all positions were confirmed with at least 35x unique coverage. The genome was annotated using Prokka v 1.7 (**Seemann, 2014**). Total reads mapping to *S. saprophyticus* from paired end-assemblies are listed in **Supplementary file 1L**.

#### *S. saprophyticus* Troy unresolved regions

There were three repetitive regions of the genome that were unable to be reliably resolved using either reference guided or iterative assembly due to the length and sequence structure of the region, and were replaced by 100 N's in the final genome sequence. These regions correspond to strain ATCC15305 positions 1810158–1810757 in SSP1741, 153694–158013 in SSP0135, and 725232–725331 between SSP0690 and SSP0691.

Regarding rRNA regions, no attempt was made to reconstruct or span these regions with iterative assembly, as the multiple copies could not be assembled accurately with short reads. All rRNA regions (genes and any flanking tRNAs) plus 200 bp of flanking sequence from the ends of the rRNA genes have been replaced by 100 N's in the final ancient genome sequence (corresponding to strain 15035 positions 743525–749373, 841186–847217, 959438–967218, 2349930–2355284, and 2301048–2312686). Any tRNAs that were not flanked by rRNA genes were included in the draft genome. The iterative assembly process established the presence of at least the same number of bordering RNA regions as

in the reference sequence, although it cannot be determined whether there is any rearrangement of the intervening regions in the Troy strain versus the reference.

### S. saprophyticus Troy plasmid, pSST1

A partially-novel plasmid was discovered during the assembly process and closed using iterative assembly. Large portions of this pSST1 plasmid have homology to the pSSAP1 (strain MS1146) and pSSP1 (strain ATCC 15305) plasmids, and a portion of the ATCC 15305 genome (similar to pSSAP1) containing *repA*, alcohol dehydrogenase, and dehydrogenase genes, which is missing in the Troy chromosomal genome (but located on the plasmid). The plasmid was annotated using Prokka v 1.7 (*Seemann, 2014*).

### The *S. saprophyticus* Troy *bla*PC1 gene encodes a functional β-lactamase

The $bla_{PC1}$ gene was submitted to SignalP (*Petersen et al., 2011*) software, which predicted an N-terminal signal peptide with a cleavage site between A32 and K33. The DNA sequence encoding mature PC1 enzyme was optimized and synthesized by IDT (Coralville, IA) before cloning into the pET-28b vector for overexpression. *Escherichia coli* BL21 chemically competent cells were transformed with pET-28b(PC1) and inoculated into LB medium containing 50 µg/mL kanamycin and grown at 37°C. Protein expression was induced with 1 mM IPTG at $OD_{600}$ 0.7 and cultures were incubated overnight at 16°C. Cells were harvested by centrifugation and cell paste from 1 L of culture expressing β-lactamase was washed with 8 mL 0.85% NaCl, resuspended in buffer containing 50 mM HEPES pH 7.5, 350 mM NaCl, and 20 mM imidazole, and then lysed by cell disruption at 20,000 PSI. Lysate was centrifuged using a Beckman JA 25.50 rotor at 20 000 RPM (48 254 x g) for 45 min at 4°C. The supernatant was applied to a 5 mL HiTrap Ni-NTA column (GE Lifesciences) at a constant flow rate of 3 mL/min. The column was washed with five column volumes of the same buffer and step gradients of increasing imidazole were used for wash and elution steps. Fractions containing purified β-lactamase, based on SDS-PAGE, were pooled and dialyzed overnight at 4°C in buffer containing 50 mM HEPES pH 7.5, 150 mM NaCl, and 20% glycerol. PC1 was determined to be >90% pure as assessed by SDS-PAGE and stored at −20°C.

Purified PC1 enzyme was used to determine kinetic parameters for nitrocefin hydrolysis (*Supplementary file 1M*). Nitrocefin was synthesized as previously reported (*Lee et al., 2005*). Enzyme (final concentration 1 nM) was added to nitrocefin in serial half dilutions (highest final concentration 320 µM) in 50 mM HEPES pH 7.5 at 30°C after a 5 s mix time. Nitrocefin hydrolysis was monitored at 490 nm and rates used to determine the kinetic parameters, $K_m$ and $k_{cat}$. All enzyme dilutions were done in 100 ng/µL bovine serum albumin and nitrocefin dilutions in 0.01% Tween 20.

## Modern comparative data generation and analyses

### Patterns of gene content and variant sharing among *S. saprophyticus* associated with distinct niches

We determined the presence and absence of known mobile genetic elements, virulence genes, and genes conferring antibiotic resistance (*Figure 4—figure supplement 5*). To gain further insights into emergence of pathogenicity, we searched for genes and genetic variants that are unique to pathogenic strains of *S. saprophyticus*. Non-synonymous variants shared between Clade P and the human UTI isolate *S. saprophyticus* 55 are listed in *Supplementary file 1O*.

Shared gene content was explored using annotated, *de novo* assemblies of *S. saprophyticus* isolates. Scripts used to automate OrthoMCL (*Li et al., 2003*) analyses and

compare gene content are available at https://github.com/tatumdmortimer/core-genome-alignment. There are 13 genes common to Clade P isolates that are absent from Clade E (**Supplementary file 1N**). However, these genes are not present in the human UTI isolate *S. saprophyticus* 55. Further studies are needed to investigate a possible role of these genes and variants in adaptation to the pathogenic niche.

## Analysis of temporal structure

No significant correlation was observed ($R^2$ = 0.037) in the regression of root-to-tip genetic distances against year of sampling using TempEst v 1.4 (**Rambaut et al., 2016**), thereby indicating that there is no temporal signal in these data, which precludes molecular clock dating analysis. Similarly, rate estimates using BEAST v 1.8 (**Drummond et al., 2012**) revealed extensive variation (95% highest posterior density: $1.4 \times 10^{-10}$–$2.9 \times 10^{-7}$ substitutions/site/year with General Time Reversible with gamma distributed site variation substitution model, lognormal relaxed clock with an exponential rate distribution, and Bayesian Skyline Plot demographic model) such that all rate estimates are necessarily unreliable.

## *S. saprophyticus* population structure

BaTS implements three statistics to test phylogenetic-trait correlation: the Fitch parsimony score (PS) (**Fitch, 1971**), the Association Index (AI) (**Wang et al., 2001**), and the maximum clade size (MC) (**Parker et al., 2008**). BaTS randomizes the taxon-trait values and uses the distribution of trees to determine null distributions of these statistics. The significance of the observed distribution of these statistics, calculated across the distribution of trees, is then determined in comparison to the null distributions. For our data, samples were coded as having either 'pathogenic' (n = 20) or non-'pathogenic' (n = 6) traits. We found that the observed distributions were significantly different from null expectations for all tests (PS: observed mean = 2.0, null mean = 5.51, p<0.001; AI: observed mean = 0.265, null mean = 1.127, p=0.007; MC of pathogenic trait: observed mean = 19.0, null mean = 5.1, p<0.001). These results indicate that pathogenic and non-pathogenic isolates in our sample are extremely structured by lineage.

