## [Decision Letter]

Thank you for submitting your article "An ancient emerging infection as a cause of maternal sepsis in Late Byzantine Troy" for consideration by *eLife*. Your article has been favorably evaluated by Richard Losick as the Senior Editor and three reviewers, including George H Perry (Reviewer #1) - who is a member of our Board of Reviewing Editors. – and Laura Weyrich (Reviewer #2).

The reviewers have discussed the reviews with one another and the Reviewing Editor has drafted this decision to help you prepare a revised submission.

Summary:

Devault et al. use DNA sequencing to describe, for the first time, urogenital infections in a woman buried ~800 years ago in Troy. The authors compare DNA present within calcified nodules (which are also analyzed morphologically) that were discovered in association with human skeletal remains to environmental, skeletal, and modern data to determine the source of the nodules and identify two bacterial species preserved within. The authors hypothesize that the infection may have been linked with pregnancy, due to some Y chromosome reads preserved within the nodule. The preservation of the *Staphylococcus saprophyticus* DNA, especially, inside the calcified nodule is astounding, providing opportunity for reconstructing a complete, high-quality genome sequence.

Essential revisions:

The reviewers have raised and discussed several major concerns that must be adequately addressed before your paper can be considered for acceptance. These necessary revisions all involve re-analysis of existing data and associated modifications to the presentation and interpretation of the results in the text, rather than generation of new data. Thus, we believe that these revisions can reasonably be completed within two months.

1) There is insufficient evidence presented in the manuscript to support the authors' conclusions about catching the "emergence" of a pathogenic human *S. saprophyticus* strain, or to determine whether this pathogen was passed from domestic animals to humans or vice versa, or even to exclude other potential scenarios in which both populations originally acquired the pathogen from the same source. This claim is made with a single cattle isolate that falls basal to the strain identified in Troy; however, other animal isolates fall within the tree. In the absence of additional cattle strains and wider sampling in general, the results should be treated with appropriate caution and interpretations tempered. Divergence date estimates generated with the field's current gold standard (BEAST) should be provided in the main text. If these estimates cannot be obtained for a technical reason or limitation then this needs to be clarified in the main text, not only the supplementary information.

1A) Note that an associated title change will likely be necessary.

2) The hypothesis of a fetal origin for the calcified nodule is intriguing, but needs further exploration. The genetic sex analyses in the paper are not rigorous to current standards, and leave more uncertainty than necessary. The authors should apply the Skoglund et al. 2013 J. Arch. Sci. (Accurate sex identification of ancient human remains using DNA shotgun sequencing) analysis to the data from both the nodule and the ulna, and then illustrate this result with a main figure, since this differential diagnosis is a key finding of the paper. The Y chromosome damage patterns presented in the supplementary information are not sufficiently robust (due to the limited number of reads) to determine if the DNA is ancient. The number of nuclear genome SNP sites covered by one or more reads in both the ulna and the nodule is likely insufficient for an assessment of relatedness. Please check and either perform the analysis or expand on the discussion of limitations in the manuscript.

3) The authors mention that extraction blank control samples were processed for enrichment, but the metagenome data for these samples are not shown. These data should be provided to further demonstrate authenticity of the results. For example, the authors could strengthen the argument that the Y chromosome and *S. saprophyticus* reads are endogenous to the sample and not the lab environment. With the high proportions of DNA mapping to *S. saprophyticus*, it is unlikely that these data reflect contaminants; however, the laboratory extraction procedure was extensive. Showing and discussing the extraction metagenome throughout the text would help to alleviate this concern.

4) The quality control assessment of whether the nodules belonged to the female whose skeletal remains they were associated has a logical flaw (easily corrected). Human mtDNA genome sequences obtained from both the nodule and from aDNA extracted from the individual's ulna were identical. The authors interpret this result as evidence that the nodule and the ulna come from the same individual. Of course, the authors elsewhere in the manuscript suggest a placental origin for the nodules, via chorioamnionitis with a male fetus. This underscores the inability for mtDNA to facilitate confident individual identifications, since it is inherited maternally without recombination. Thus, the authors should be more circumspect in their identification based on this evidence. If this is DNA from a male fetus, then all evidence is indeed pointing to an association with the female whose skeletal remains were recovered, but please ensure precision when discussing this evidence.

5) There is currently no description provided of the osteological materials and the methods and results leading to the conclusion that the skeletal remains are from a likely female individual, of ~30 years of age. This needs to be remedied, as these designations are critical for the interpretation.

6) All data must be provided via appropriate repositories, with clear descriptions of accession numbers for each dataset. Sequence reads from all experiments (and ideally intermediate processing files, and alignments, e.g., through Dryad), including the metagenomic datasets and the raw data from the bone (i.e., not simply the genome sequences), the SEM data, XRD data, etc., associated with this work all need to be thoroughly curated and made available.

---

## [Author Response]

*Essential revisions:*

*The reviewers have raised and discussed several major concerns that must be adequately addressed before your paper can be considered for acceptance. These necessary revisions all involve re-analysis of existing data and associated modifications to the presentation and interpretation of the results in the text, rather than generation of new data. Thus, we believe that these revisions can reasonably be completed within two months.*

*1) There is insufficient evidence presented in the manuscript to support the authors' conclusions about catching the "emergence" of a pathogenic human S. saprophyticus strain, or to determine whether this pathogen was passed from domestic animals to humans or vice versa, or even to exclude other potential scenarios in which both populations originally acquired the pathogen from the same source. This claim is made with a single cattle isolate that falls basal to the strain identified in Troy; however, other animal isolates fall within the tree. In the absence of additional cattle strains and wider sampling in general, the results should be treated with appropriate caution and interpretations tempered. Divergence date estimates generated with the field's current gold standard (BEAST) should be provided in the main text. If these estimates cannot be obtained for a technical reason or limitation then this needs to be clarified in the main text, not only the supplementary information.*

*1A) Note that an associated title change will likely be necessary.*

We appreciate the reviewers’ comments that we have insufficient evidence to identify the emergence of a human pathogenic lineage of *S. saprophyticus*. We have therefore removed all references to this concept and changed the title to “A molecular portrait of maternal sepsis from Byzantine Troy”.

We attempted to estimate the nucleotide substitution rate for *S. saprophyticus* but were unable to generate precise estimates due to a striking lack of temporal signal in the data. We have added a section to the main text that describes these attempts and our hypotheses about potential reasons we failed to observe any temporal signal.

Upon re-review of the supplementary material, we decided that the rate analyses for *Gardnerella vaginalis* were based on a sample of insufficient size and we have removed this paragraph.

*2) The hypothesis of a fetal origin for the calcified nodule is intriguing, but needs further exploration. The genetic sex analyses in the paper are not rigorous to current standards, and leave more uncertainty than necessary. The authors should apply the Skoglund et al. 2013 J. Arch. Sci. (Accurate sex identification of ancient human remains using DNA shotgun sequencing) analysis to the data from both the nodule and the ulna, and then illustrate this result with a main figure, since this differential diagnosis is a key finding of the paper.*

Our hypothesis about the origin of the nodules was not well described in the original manuscript: we don’t posit a fetal origin for the nodules, rather we hypothesize that they are of placental origin. Placental tissue is of both fetal and maternal origin, and the inflammatory response to infection of the placenta and fetal membranes involves cells from mother and fetus. Our observations of the nodules are consistent with their having an origin as placental abscesses consisting of bacterial cells, maternal inflammatory cells with a minority component of fetal inflammatory cells. We have amended the description in the main text to make this logic more explicit and clear to the reader.

To address comments regarding appropriate sexing of the remains we have applied the methodology of Skoglund et al. This assigns a female sex to the nodules, with high certainty, reflecting a majority maternal origin/signal for the inflammatory cells in the abscess. We have added this detail to the main text and the supplementary material with an associated table. We don’t however think that a figure is necessary to convey this information and hope the reviewer concurs.

We also agree with the reviewer that there are insufficient data from nuclear SNPs for an assessment of relatedness and thus feel that a figure would unduly draw attention to this aspect of the paper. We have modified the text accordingly (re: sexing).

“We attempted to sex the remains (i.e. calcified nodules, ulna and their associated blanks as well as the surrounding sediments and blanks) using the method developed and published by Skoglund et al. (2013). […] Importantly, blanks, also with low read counts (between 1 and 35) are labelled as ‘consistent with XX’ however with 95% confidence intervals (at those levels) of 0.”

*The Y chromosome damage patterns presented in the supplementary information are not sufficiently robust (due to the limited number of reads) to determine if the DNA is ancient. The number of nuclear genome SNP sites covered by one or more reads in both the ulna and the nodule is likely insufficient for an assessment of relatedness. Please check and either perform the analysis or expand on the discussion of limitations in the manuscript.*

The number of reads mapping to the Y in our nodule libraries is low (884) but not (as we have shown) insignificant, when compared to all necessary controls (sediment, blanks, etc.).

While the signal of deamination (as C —> T and G—> A) is present at the termini (-1 to – 14bp), it is indeed low. This is due to the fact that the majority of Y reads come from the more deeply sequenced, UDG treated libraries.

In summary, we feel that the Y- chromosomal FLDs’ low but none-the-less present damage signal, their independent identification in the Skoglund et al. 2013 method (at the reviewers’ suggestion), and our Fisher’s exact test, suggest their authenticity more so than their identification as spurious low level contamination not found in equally sequenced bone or sedimentary remains. We hope the reviewers agree.

*3) The authors mention that extraction blank control samples were processed for enrichment, but the metagenome data for these samples are not shown. These data should be provided to further demonstrate authenticity of the results. For example, the authors could strengthen the argument that the Y chromosome and S. saprophyticus reads are endogenous to the sample and not the lab environment. With the high proportions of DNA mapping to S. saprophyticus, it is unlikely that these data reflect contaminants; however, the laboratory extraction procedure was extensive. Showing and discussing the extraction metagenome throughout the text would help to alleviate this concern.*

We apologize if this was not clear, there are a lot of data spread across various tables. The results the reviewer is requesting are indeed listed in [Supplementary-material SD10-data]. This table is (as quoted in the supplementary file) a “Summary of unique shotgun reads (from nodules, ulna, sediment and associated blanks) of minimum length 35bp and minimum mapping quality of 30 (or greater), mapping to Staphylococcus saprophyticus and Gardnerella vaginalis. Nodule pooled =Nod2-UDG + Nod1_1h-nonU + Nod2-nonU + Nod1_1h-UDG).”

As the reviewers will see, there are low level reads in the blanks that do map to *S. saprophyticus* (0-53 reads, the highest being the sediment) and *G. vaginalis* (0-9). In both cases, when compared with the nodules these represent minute fractions of what’s detected in the nodules and thus unlikely to be contamination. We did, in addition, perform metagenomic analysis on all reads found in the blanks. Reads (>100) that could be identified at the sequence level or the species/strain level from all blank extracts were removed from final files used in the PCA analysis.

*4) The quality control assessment of whether the nodules belonged to the female whose skeletal remains they were associated has a logical flaw (easily corrected). Human mtDNA genome sequences obtained from both the nodule and from aDNA extracted from the individual's ulna were identical. The authors interpret this result as evidence that the nodule and the ulna come from the same individual. Of course, the authors elsewhere in the manuscript suggest a placental origin for the nodules, via chorioamnionitis with a male fetus. This underscores the inability for mtDNA to facilitate confident individual identifications, since it is inherited maternally without recombination. Thus, the authors should be more circumspect in their identification based on this evidence. If this is DNA from a male fetus, then all evidence is indeed pointing to an association with the female whose skeletal remains were recovered, but please ensure precision when discussing this evidence.*

As noted in the response to question 2 above, we believe that the nodules are of both maternal and fetal origin. As the reviewers point out, mitochondrial sequences of maternal inflammatory cells, fetal inflammatory cells and maternal bone cells should be identical. Our finding of identical mitochondrial haplotypes in the ulna and nodules is consistent with this scenario.

We feel that the most parsimonious explanation of our observation of identical haplotypes in the nodules and ulna is that we have enriched and sequenced the identical woman’s mitogenome, with a possible minority component from her fetus. However, there is no precise way, without autosomal SNPs, to be 100% certain. For this reason, we have modified the following sentence in the main text to read:

“The nodule and the ulna share the identical mitochondrial haplotype (Supplementary file 5), suggesting that they likely stem from the same individual or a maternal relative.”

*5) There is currently no description provided of the osteological materials and the methods and results leading to the conclusion that the skeletal remains are from a likely female individual, of ~30 years of age. This needs to be remedied, as these designations are critical for the interpretation.*

Thank you for pointing this out. We have added a detailed description of the osteological materials and methods used for sexing and age estimation of the woman at Troy’s remains (1^st^ section of the supplement).

*6) All data must be provided via appropriate repositories, with clear descriptions of accession numbers for each dataset. Sequence reads from all experiments (and ideally intermediate processing files, and alignments, e.g., through Dryad), including the metagenomic datasets and the raw data from the bone (i.e., not simply the genome sequences), the SEM data, XRD data, etc., associated with this work all need to be thoroughly curated and made available.*

The sequencing data have been deposited under the BioProject IDs PRJNA352403 and PRJNA352376. Other data supporting the paper are included as supplementary material.